# IMPROVING OBJECT-CENTRIC LEARNING WITH QUERY OPTIMIZATION

**Baoxiong Jia**[1,3†*]**, Yu Liu**[2,3†*]**, Siyuan Huang**[3]
[1]UCLA  [2]Tsinghua University  [3] National Key Laboratory of General Artificial Intelligence, BIGAI

## ABSTRACT

The ability to decompose complex natural scenes into meaningful object-centric abstractions lies at the core of human perception and reasoning. In the recent culmination of unsupervised object-centric learning, the Slot-Attention module has played an important role with its simple yet effective design and fostered many powerful variants. These methods, however, have been exceedingly difficult to train without supervision and are ambiguous in the notion of object, especially for complex natural scenes. In this paper, we propose to address these issues by investigating the potential of learnable queries as initializations for Slot-Attention learning, uniting it with efforts from existing attempts on improving Slot-Attention learning with bi-level optimization. With simple code adjustments on Slot-Attention, our model, Bi-level Optimized Query Slot Attention, achieves state-of-the-art results on 3 challenging synthetic and 7 complex real-world datasets in unsupervised image segmentation and reconstruction, outperforming previous baselines by a large margin. We provide thorough ablative studies to validate the necessity and effectiveness of our design. Additionally, our model exhibits great potential for concept binding and zero-shot learning. Our work is made publicly available at https://bo-qsa.github.io.

## 1 INTRODUCTION

Objects, and their interactions, are the foundations of human cognition (Spelke & Kinzler, 2007). The endowment on making abstractions from perception and organizing them systematically empowers humans the ability to accomplish and generalize across a broad range of tasks, such as scene modeling (Bear et al., 2020), visual reasoning (Yi et al., 2020), and simulating interactions (Bear et al., 2020). The key to such success lies in the emergence of symbol-like mental representations of object concepts (Whitehead, 1928). However, important as it is, disentangling object-centric concepts from visual stimuli is an exceedingly difficult task to accomplish with limited supervision (Greff et al., 2020) and requires proper inductive biases (Schölkopf et al., 2021).

Motivated by the development of symbolic thought in human cognition, slot-based representations, instance (Greff et al., 2017; 2019; Locatello et al., 2020), sequential (Gregor et al., 2015; Burgess et al., 2019; Engelcke et al., 2021; Goyal et al., 2021), or spatial (Crawford & Pineau, 2019; Lin et al., 2020; Jiang et al., 2019), have been the key inductive bias to recent advances in unsupervised object-centric learning. Among them, the Slot-Attention module has received tremendous focus given its simple yet effective design (Locatello et al., 2020). By leveraging the iterative attention mechanism, Slot-Attention learns to compete between slots for explaining parts of the input, exhibiting a soft-clustering effect on visual signals. It is later proven to be more memory and training efficient as a plug-and-play module for unsupervised object-centric learning (Locatello et al., 2020) and fostered powerful variants in understanding images (Singh et al., 2021; Xu et al., 2022), 3D scenes (Yu et al., 2022; Sajjadi et al., 2022a) and videos (Kipf et al., 2022; Elsayed et al., 2022; Singh et al., 2022).

However, as revealed by recent studies, the Slot-Attention module comes with innate discrepancies for object-centric representation learning. First, with slots randomly initialized each time, the object-centric representations obtained by these models do not necessarily bind to object concepts (Kipf et al., 2022). Intuitively, such randomness leads to undesired scenarios where slots with similar

---

*Equal contribution. †Work done during internship at BIGAI.

initializations compete for objects on different images. Such randomness challenges the iterative refinement procedure as it now needs to project sets of potentially similar representations to independent constituents of the input. As discovered by Chang et al. (2022), differentiating through such recurrences contributes to various training instabilities with growing spectral norm of Slot-Attention weights. This leads to the second and perhaps least desired property of Slot-Attention; it relies heavily on hyper-parameter tuning, including gradient clipping, learning rate warm-up, *etc.*, and further hurts the flexibility of Slot-Attention in adapting to broader applications with more complex signals.

To this end, we propose an extension of the Slot-Attention module, Bi-level Optimized Query Slot Attention (BO-QSA), to tackle the aforementioned problems. First, we follow the bi-level optimization framework proposed by Chang et al. (2022) for easing the training difficulty in Slot-Attention. More importantly, instead of sampling from a learnable Gaussian distribution, we propose to directly learn the slot initializations as queries. With these learnable representations, we eliminate the ambiguous competitions between slots and provide a better chance for them to bind to specific object concepts. We improve the training of query-initialized Slot-Attention with a straight-through gradient estimator (STE) by connecting our method with first-order approaches (Finn et al., 2017; Nichol & Schulman, 2018; Geng et al., 2021) in solving bi-level optimization problems. The experimental results show that the proposed BO-QSA can achieve state-of-the-art results on both synthetic and real-world image datasets with simple code adjustments to the original Slot-Attention module.

With our model significantly outperforming previous methods in both synthetic and real domains, we provide thorough ablative studies demonstrating the effectiveness of our model design. We later show that our BO-QSA possesses the potential of binding object concepts to slots. To validate this potential, we design zero-shot transfer learning experiments to show the generalization power of our model on unsupervised object-centric learning. As the experiments suggest (see Sec. 5), our model could potentially be a principle approach for unsupervised object-centric learning and serve as a general plug-and-play module for a broader range of modalities where variants of Slot-Attention prosper. We hope these efforts can help foster new insights in the field of object-centric learning.

**Contributions**    In summary, our main contributions are three-fold:

- We propose BO-QSA, a query-initialized Slot-Attention model that unites straight-through gradient updates to learnable queries with methods on improving Slot-Attention with bi-level optimization.
- We show that, with simple code adjustments on Slot-Attention, the proposed BO-QSA achieves state-of-the-art results on several challenging synthetic and real-world image benchmarks, outperforming previous methods by a large margin.
- We show the potential of our BO-QSA being a better approach to concept binding and learning generalizable representations with qualitative results and zero-shot transfer learning experiments.

## 2   PRELIMINARIES

### 2.1   OBJECT-CENTRIC REPRESENTATION LEARNING WITH SLOT-ATTENTION

Slot-Attention (Locatello et al., 2020) takes a set of $N$ input feature vectors $\mathbf{x} \in \mathbb{R}^{N \times D_{\text{input}}}$ and maps them to a set of $K$ output vectors (*i.e.*, slots) $\mathbf{s} \in \mathbb{R}^{K \times D_{\text{slots}}}$. It leverages an iterative attention mechanism to first map inputs and slots to the same dimension $D$ with linear transformations $k(\cdot)$, $q(\cdot)$ and $v(\cdot)$ parameterized by $\phi^{\text{attn}}$. At each iteration, the slots compete to explain part of the visual input by computing the attention matrix $\boldsymbol{A}$ with softmax function over slots and updating slots with the weighted average of visual values:

$$\tilde{\mathbf{s}} = f_{\phi^{\text{attn}}}(\mathbf{s}, \mathbf{x}) = \left( \frac{A_{i,j}}{\sum_{l=1}^{N} A_{l,j}} \right)^{\top} \cdot v(\mathbf{x}) \quad \text{where} \quad \boldsymbol{A} = \text{softmax}\left( \frac{k(\mathbf{x}) \cdot q(\mathbf{s})^{\top}}{\sqrt{D}} \right) \in \mathbb{R}^{N \times K}.$$

The slots are initialized from a learnable Gaussian distribution with mean $\boldsymbol{\mu}$ and variance $\boldsymbol{\sigma}$. They are refined iteratively within the Slot-Attention module by passing the updates into a Gated Recurrent Unit (GRU) (Cho et al., 2014) and MLP parameterized by $\phi^{\text{update}}$ for $T$ iterations:

$$\mathbf{s}^{(t+1)} = h_{\phi^{\text{update}}}(\mathbf{s}^{(t)}, \tilde{\mathbf{s}}^{(t)}), \quad \mathbf{s}^0 \sim \mathcal{N}(\boldsymbol{\mu}, \text{diag}(\boldsymbol{\sigma})), \quad \hat{\mathbf{s}} = \mathbf{s}^{(T)}. \tag{1}$$

The final prediction $\hat{\mathbf{s}}$ can be treated as the learned object-centric representation w.r.t. to input features $\mathbf{x}$. In the image domain, we take as input a set of images $\boldsymbol{I}$ and encode them with $f_{\phi^{\text{enc}}}$ to obtain

features $\mathbf{x} \in \mathbb{R}^{HW \times D_{\text{input}}}$. After obtaining $\hat{\mathbf{s}}$ through the iterative refinement procedure with $h_{\phi^{\text{update}}}$, images could be decoded from these object-centric representations with a mixture-based decoder or autoregressive transformer-based decoder. We refer the readers to Appendix A.1 for details on different decoder designs and their ways of visualizing learned object concepts.

## 2.2 IMPROVING SLOT-ATTENTION WITH BI-LEVEL OPTIMIZATION

The problem of bi-level optimization embeds the optimization of an inner objective within the outer objective. Normally, a bi-level optimization problem can be formulated as:

$$\min_{\theta, \phi} f(\theta, \phi) \quad s.t. \quad \theta \in \arg\min_{\theta'} g(\theta', \phi), \tag{2}$$

where we call $f(\theta, \phi)$ the outer objective function and $g(\theta, \phi)$ the inner objective function. To jointly optimize both objectives w.r.t. parameters $\theta$ and $\phi$, a straightforward approach to solving Eq. (2) is to represent the inner solution of $\theta$ as a function of $\phi$, $i.e.$, $\theta^*(\phi) = \arg\min_{\theta'} g(\theta', \phi)$. Then we can optimize the outer objective with gradient descent by approximating $\nabla_\phi f(\theta^*(\phi), \phi)$ as a function of $\phi$. When the inner optimization objective could be solved by a fixed point iteration $\theta = F_\phi(\theta)$ (Amos & Kolter, 2017; Bai et al., 2019), the bi-level optimization problem could be solved by

$$\frac{\partial f(\theta^*(\phi), \phi)}{\partial \phi} = \frac{\partial f(\theta^*(\phi), \phi)}{\partial \theta^*} \cdot \sum_{i=0}^{\infty} \left( \frac{\partial F_\phi(\theta^*)}{\partial \theta^*} \right)^i \cdot \frac{\partial F_\phi(\theta^*)}{\partial \phi}. \tag{3}$$

For efficiency concerns, recent methods often use the first-order approximation of the infinite Neumann's series (Shaban et al., 2019; Geng et al., 2021) for updating $\phi$. Given that Slot-Attention is, in essence, an iterative refinement method that falls into the same framework, Chang et al. (2022) adapted this technique to improve Slot-Attention training and obtained significant improvement both in model performance and training stability. We provide more discussions on this in Sec. 3.2 and also other bi-level optimization methods for approximating $\nabla_\phi f(\theta^*(\phi), \phi)$ in Appendix A.2.

## 3 METHOD

### 3.1 QUERY SLOT ATTENTION

As mentioned in Sec. 1, the Slot-Attention module adopts a random initialization of slots and conducts iterative refinement to obtain object-centric representations $\hat{\mathbf{s}}$ as in Eq. (1). However, as argued by Kipf et al. (2022), such random initializations provide no hint on the notion of object and no means for controllably probing concepts from the model. As shown by Chang et al. (2022), this random initialization plays a minimal role and could be detached from training. This indicates that the estimation of $\hat{\mathbf{s}}$ relies heavily on the task-specific iterative refining of slots over data, leaving a limited possibility for slots to bind to specific concepts and be leveraged as generalizable representations.

To address this issue, we focus on the Query Slot Attention (QSA), which initializes the slots in the Slot-Attention module with learnable queries $\mathbf{s}_0 = \phi^{\text{init}}$. Such a design is motivated by the success of recent query-based networks (Van Den Oord et al., 2017; Jaegle et al., 2021b). It facilitates an object-centric model to learn general symbolic-like representations that could be quickly adapted by refining over task-specific requirements, as discussed in Sec. 1 and Kipf et al. (2022). Meanwhile, in contrast to the use of learnable queries in other encoder-decoder structures ($e.g.$ discrete VAE (dVAE)), the slot initializations $\mathbf{s}_0$ are not necessarily required to encode image features since they were designed for separating them. This resembles recent discoveries in query networks (Carion et al., 2020; Yang et al., 2021) where queries could be generalizable probes for input properties. Despite the good properties and potentials QSA presents, it is shown detrimental to initialize slots independently in Slot-Attention under unsupervised settings (Locatello et al., 2020).

### 3.2 RETHINKING BI-LEVEL OPTIMIZATION METHODS FOR QUERY SLOT ATTENTION

To improve the learning of QSA, we rewind to the idea of improving the learning of the vanilla Slot-Attention module with bi-level optimization (Chang et al., 2022). Under this formulation, Slot-Attention could be treated as solving the following objectives:

$$\min_{\mathbf{s}, \Phi} \sum_{i=1}^{M} \mathcal{L}(\mathbf{x}_i, \mathbf{s}_i, \Phi) \quad s.t. \quad \mathbf{s}_i^* = \arg\min_{\mathbf{s}} \mathcal{L}_{\text{cluster}}(\mathbf{x}_i, \mathbf{s}, \Phi), \tag{4}$$

where $\mathbf{x}_i$ and $\mathbf{s}_i$ denote the input feature from the $i$-th image and its corresponding slots, and $\Phi = \{\phi^{\text{init}}, \phi^{\text{attn}}, \phi^{\text{update}}\}$ denotes parameters for assigning input features $\mathbf{x}$ to different slots. Under this setting, the outer objective $\mathcal{L}$ is usually a reconstruction objective and the inner objective could be viewed as a soft-clustering objective (Locatello et al., 2020). Next, the inner objective is solved by iterative refinement, which could be formulated as solving for fixed-points (Chang et al., 2022) of

$$\mathbf{s} = h_{\phi^{\text{update}}}(\mathbf{s}, \tilde{\mathbf{s}}) = h_{\phi^{\text{update}}}(\mathbf{s}, f_{\phi^{\text{attn}}}(\mathbf{s}, \mathbf{x})) = F_{\Phi}(\mathbf{s}, \mathbf{x}), \tag{5}$$

where $F_{\Phi}(\cdot, \cdot)$ is an fixed-point operation. As introduced by Chang et al. (2022) in Implicit Slot-Attention (I-SA), with Eq. (3), the instabilities through the iterative updates could be avoided by detaching gradients, treating slots in the final iteration as an approximation of $\mathbf{s}_i^*$, and computing first-order gradient approximations for updating $\Phi$ with $\mathbf{s}_i^*$. However, we demonstrate in Tab. 7 that this design is only beneficial for randomly initialized slots and detrimental for query-initialized Slot-Attention architectures since it relies heavily on the good approximation of the solution to the inner objective. With no randomness in slot initializations or gradient during training, starting from a fixed set of initialization points puts challenges on the learning of Slot-Attention update $F_{\Phi}$ as it will be difficult to provide a good approximation of $s_i^*$ with only a fixed number of iterations (see in Appendix B.2). This urges the need for information flow to the slot initialization queries.

### 3.3 BI-LEVEL OPTIMIZED QUERY SLOT ATTENTION

We propose BO-QSA to address the learning problem of QSA. As shown in Algorithm 1, we initialize slots with learnable queries in BO-QSA and perform $T$ steps of Slot-Attention update to obtain an approximation of $\mathbf{s}_i^*$. These near-optimal solutions of the inner objective are passed into one additional Slot-Attention step where gradients to all previous iterations are detached. In contrary to I-SA, we use a STE (Bengio et al., 2013; Van Den Oord et al., 2017) to backpropagate gradients and also to slot initialization queries. Such

---

**Algorithm 1: BO-QSA**

**Input:** input features `input`, learnable queries `init`, number of iterations $T$
**Output:** object-centric representation `slots`
**Modules:** stop gradient module SG($\cdot$), slot attention module SA($\cdot, \cdot$)

`slots = init`
**for** $t = 1, \cdots, T$ **do**
  |   `slots = SA(slots, inputs)`
`slots = SG(slots) + init - SG(init)`
`slots = SA(slots, inputs)`
**return** `slots`

---

designs help find good starting points for the inner optimization problem on clustering, alleviating the problem of bi-level optimization with QSA mentioned in Sec. 3.2. Similar to dVAE, the STE adds bias to the gradient of the initialization queries. However, since these learnable queries are meant for disentangling image features, they do not have to maintain information about the approximated $\mathbf{s}^*$. Such bias could lead to learned queries which are better pivots for separating different image features, similar to anchors, or filter queries learned for different tasks (Carion et al., 2020; Zhang et al., 2021). Note that we do not add constraints on the consistency between $\mathbf{s}_0$ and $\hat{\mathbf{s}}$ (e.g. $||sg(\hat{\mathbf{s}}) - \mathbf{s}_0||^2$) as done in dVAE since we find such constraints lead to a mean-representation of datasets that forbids better concept binding (see in Appendix B.3). As shown in Tab. 7 and Fig. 3, our learned slot initialization queries do fulfill this goal by providing a more separable initialization space and can significantly facilitate model learning.

## 4 RELATED WORK

**Unsupervised Object-Centric Learning** Our work falls into the recent line of research on unsupervised object-centric learning on images (Greff et al., 2016; Eslami et al., 2016; Greff et al., 2017; 2019; Burgess et al., 2019; Crawford & Pineau, 2019; Engelcke et al., 2020; Lin et al., 2020; Bear et al., 2020; Locatello et al., 2020; Zoran et al., 2021). A thorough review and discussion on this type of method can be found in Greff et al. (2020). One critical issue of these methods is on handling complex natural scenes. Singh et al. (2021); Lamb et al. (2021) leverages a transformer-based decoder with Slot-Attention for addressing this problem. Similar attempts have also been made by exploiting self-supervised contrastive learning (Choudhury et al., 2021; Caron et al., 2021; Wang et al., 2022; Hénaff et al., 2022) and energy-based models (Du et al., 2021; Yu et al., 2022). Our work builds upon Slot-Attention by extending it with learnable queries and a novel optimization method for learning. Our compelling experimental suggests our model could potentially serve as a general plug-and-play module for a wider range of modalities where variants of Slot-Attention prosper (Kipf et al., 2022; Elsayed et al., 2022; Singh et al., 2022; Yu et al., 2022; Sajjadi et al., 2022a;b).

**Query Networks**    Sets of latent queries are commonly used in neural networks. These methods leverage permutation equivariant network modules (*e.g.* GNNs (Scarselli et al., 2008) and attention modules (Vaswani et al., 2017)) in model design for solving set-related tasks such as clustering (Lee et al., 2019), outlier detection (Zaheer et al., 2017; Zhang et al., 2019), *etc*. These learned latent queries have been shown to have good potential as features for tasks like contrastive learning (Caron et al., 2020), object detection (Carion et al., 2020), and data compression (Jaegle et al., 2021a;b). In contrast to the recent success of query networks in supervised or weakly-supervised learning (Carion et al., 2020; Zhang et al., 2021; Kipf et al., 2022; Elsayed et al., 2022; Xu et al., 2022), Locatello et al. (2020) demonstrates the detrimental effect of using independently initialized slots in Slot-Attention learning. However, we show that our BO-QSA method successfully overcomes this issue and generalizes the success of query networks to the domain of unsupervised object-centric learning.

**Bi-level Optimization**    Our work is closely related to bi-level optimization methods with iterative fixed update rules for solving the inner objective. Specifically, methods are designed with implicit differentiation (Amos & Kolter, 2017; Bai et al., 2019) to stabilize the iterative update procedure. Similar formulations are also found when combined with meta-learning where Madan et al. (2021) train queries through recurrence in a meta-learning fashion and Rajeswaran et al. (2019) provides a unified view of the optimization problem with implicit gradients. Concurrent work from Chang et al. (2022) formulate the Slot-Attention learning from an implicit gradient perspective with gradient stopping derived from first-order hyper-gradient methods (Geng et al., 2021). However, they ignore the important role of slot initializations in generalization and concept binding. As our experiments suggest, such gradient-stopping methods do not guarantee superior performance compared to the original Slot-Attention. We leave the details to Sec. 5.3 for an in-depth discussion.

## 5    EXPERIMENTS

In this section, we aim to address the following questions with our experimental results:

- How good is our proposed BO-QSA on both synthetic and complex natural scenes?
- How important is the query and the optimization method in BO-QSA?
- Does BO-QSA possess the potential for concept binding and zero-shot transfer?

We provide details in the following sections with thorough comparative and ablative experiments and leave the details on model implementation and hyperparameter selection to Appendix A.3. Here we clarify the datasets and metrics selected for evaluating our model on each domain:

**Synthetic Domain**    For the synthetic domain, we select three well-established challenging multi-object datasets Shapestacks (Groth et al., 2018), ObjectsRoom (Kabra et al., 2019), and CLEVRTEX for evaluating our BO-QSA model. Specifically, we consider three metrics to evaluate the quality of object segmentation and reconstruction. Adjusted Rand Index (ARI) (Hubert & Arabie, 1985) and Mean Segmentation Covering (MSC) (Engelcke et al., 2020) for segmentation and Mean Squared Error (MSE) for reconstruction. Following the evaluation setting of recent works, we report the first two segmentation metrics over foreground objects (ARI-FG and MSC-FG). Additionally, we conduct extra experiments on more datasets and leave the discussion to Appendix B.1.

**Real-world Images**    For the real image domain, we use two tasks (1) unsupervised foreground extraction and (2) unsupervised multi-object segmentation for evaluating our method. Specifically, we select Stanford Dogs (Khosla et al., 2011), Stanford Cars (Krause et al., 2013), CUB200 Birds (Welinder et al., 2010), and Flowers (Nilsback & Zisserman, 2010) as our benchmarking datasets for foreground extraction and YCB (Calli et al., 2017), ScanNet (Dai et al., 2017), COCO (Lin et al., 2014) proposed by Yang & Yang (2022) for multi-object segmentation. We use mean Intersection over Union (mIoU) and Dice as metrics for evaluating the quality of foreground extraction and use the evaluation metrics adopted by Yang & Yang (2022) for multi-object segmentation.

### 5.1    OBJECT DISCOVERY ON SYNTHETIC DATASETS

**Experimental Setup**    We explore our proposed BO-QSA with two types of decoder designs, mixture-based and transformer-based, as discussed in Sec. 2.1 and Appendix A.1. We follow the decoder architecture in Slot-Attention (Locatello et al., 2020) for mixture-based decoders and

Table 1: Multi-object segmentation results on ShapeStacks and ObjectsRoom. We report ARI-FG and MSC-FG of all models with (mean $\pm$ variance) across 3 experiment trials. We visualize the best results in bold.

| Model | ShapeStacks | | ObjectsRoom | |
|---|---|---|---|---|
| | ↑ ARI-FG | ↑ MSC-FG | ↑ ARI-FG | ↑ MSC-FG |
| MONet-G (Burgess et al., 2019) | 0.70$\pm$0.04 | 0.57$\pm$0.12 | 0.54$\pm$0.00 | 0.33$\pm$0.01 |
| GENESIS (Engelcke et al., 2020) | 0.70$\pm$0.05 | 0.67$\pm$0.02 | 0.63$\pm$0.03 | 0.53$\pm$0.07 |
| Slot-Attention (Locatello et al., 2020) | 0.76$\pm$0.01 | 0.70$\pm$0.05 | 0.79$\pm$0.02 | 0.64$\pm$0.13 |
| GENSIS-V2 (Engelcke et al., 2021) | 0.81$\pm$0.01 | 0.67$\pm$0.01 | 0.86$\pm$0.01 | 0.59$\pm$0.01 |
| SLATE (Singh et al., 2021) | 0.65$\pm$0.03 | 0.63$\pm$0.05 | 0.57$\pm$0.03 | 0.30$\pm$0.03 |
| I-SA (Chang et al., 2022) | 0.90$\pm$0.02 | 0.85$\pm$0.03 | 0.85$\pm$0.01 | 0.76$\pm$0.04 |
| Ours (transformer) | 0.68$\pm$0.02 | 0.70$\pm$0.02 | 0.68$\pm$0.03 | 0.72$\pm$0.03 |
| Ours (mixture) | **0.93$\pm$0.01** | **0.89$\pm$0.00** | **0.87$\pm$0.03** | **0.80$\pm$0.02** |

Table 2: Multi-object segmentation results on CLEVRTEX. We report ARI-FG (%) and MSE of all models in the form of (mean $\pm$ variance) across 3 experiment trials. We visualize the best results in bold.

| Model | CLEVRTEX-FULL | | CLEVRTEX-OOD | | CLEVRTEX-CAMO | |
|---|---|---|---|---|---|---|
| | ↑ ARI-FG | ↓ MSE | ↑ ARI-FG | ↓ MSE | ↑ ARI-FG | ↓ MSE |
| MONet (Burgess et al., 2019) | 19.78$\pm$1.02 | **146$\pm$7** | 37.29$\pm$1.04 | 409$\pm$3 | 31.52$\pm$0.87 | 265$\pm$1 |
| Slot-Attention (Locatello et al., 2020) | 62.40$\pm$2.33 | 254$\pm$8 | 58.45$\pm$1.87 | 487$\pm$16 | 57.54$\pm$1.01 | **215$\pm$7** |
| GENSIS-V2 (Engelcke et al., 2021) | 31.19$\pm$12.41 | 315$\pm$106 | 29.04$\pm$11.23 | 539$\pm$147 | 29.60$\pm$12.84 | 278$\pm$75 |
| DTI (Monnier et al., 2021) | 79.90$\pm$1.37 | 438$\pm$22 | 73.67$\pm$0.98 | 590$\pm$4 | **72.90$\pm$1.89** | 377$\pm$17 |
| I-SA (Chang et al., 2022) | 78.96$\pm$3.88 | 280$\pm$8 | 83.71$\pm$0.88 | **241$\pm$4** | 57.20$\pm$13.28 | 295$\pm$30 |
| Ours (mixture) | **80.47$\pm$2.49** | 268$\pm$2 | **86.50$\pm$0.19** | 265$\pm$25 | 63.71$\pm$6.11 | 280$\pm$7 |

SLATE (Singh et al., 2021) for transformer-based decoders. For both types of models, we use the Slot-Attention module with a CNN image encoder and initialize slots with learnable embeddings.

**Results** We report multi-object segmentation results on synthetic datasets in Tab. 1 and visualize qualitative results in Fig. 1. As shown in Tab. 1, our BO-QSA achieves the state-of-the-art results with large improvements over previous object-centric learning methods on all metrics in ShapeStacks and ObjectsRoom. We also observe more stable model performance, *i.e.* smaller variances in results, across different trials of experiments. Our model with mixture-based decoders obtains the best overall performance on all datasets. More specifically, our mixture-based BO-QSA significantly outperforms the vanilla Slot-Attention model ($\sim$15%) with minimal architectural differences. This validates the importance of the learnable queries and our optimization method. We will continue this discussion in Sec. 5.3. As shown in Tab. 2, our model also achieves state-of-the-art results on the unsupervised object segmentation task in CLEVRTEX with consistent improvement over Slot-Attention on the CAMO and OOD generalization split. Interestingly, our model (1) shows larger reconstruction errors, (2) generalizes well in out-of-distribution scenarios, and (3) shows marginal improvement in camouflaged images. We attribute (1) and (3) to the simple architecture of encoders/decoders currently adopted and provide insights on (2) in Sec. 5.4.

**Mixture-based vs. Transformer-based Decoder** We observe inferior segmentation but superior reconstruction performance of transformer-based variants of Slot-Attention on synthetic datasets. Specifically, we compare the MSE of models on ShapeStacks and ObjectsRoom. As shown in Tab. 3, transformer-based methods provide better reconstruction results. We attribute the low segmentation performance

Table 3: Reconstruction results on ShapeStacks and ObjectsRoom (MSE↓). We compare mixture-based and transformer-based decoder designs.

| Model | ShapeStacks | ObjectsRoom |
|---|---|---|
| Slot-Attention (mixture) | 80.8 | 20.4 |
| ours (mixture) | **72.0** | **8.1** |
| SLATE (transformer) | 52.3 | 16.3 |
| ours (transformer) | **49.3** | **14.7** |

to mask prediction in these methods, which relies on the attention matrix computed over input features. This leads to coarse object masks as a result of image tokenization. Nonetheless, we observe consistent improvement by applying our slot encoder to both mixture and transformer decoders.

## 5.2 OBJECT DISCOVERY ON REAL DATASETS

**Experimental Setup** For real-world experiments, we use the same slot encoder design used in Sec. 5.1 with a 4-layer CNN image encoder and initialize slots with learnable queries. For

Table 4: Unsupervised multi-object segmentation results on YCB, ScanNet, and COCO variant proposed by Yang & Yang (2022). We use the same evaluation metrics as in Yang & Yang (2022) and report all models' results with (mean (variance)) over 3 experiment trials. We visualize the best results in bold.

| Model | YCB (AP / PQ / Pre / Rec) ↑ | ScanNet (AP / PQ / Pre / Rec) ↑ | COCO (AP / PQ / Pre / Rec) ↑ |
|---|---|---|---|
| AIR (Eslami et al., 2016) | 0.0 (0.1) / 0.6 (0.3) / 1.1 (0.4) / 0.8 (0.2) | 2.7 (1.4) / 6.3 (1.7) / 15.6 (2.8) / 7.3 (1.6) | 2.7 (0.1) / 6.7 (0.5) / 14.3 (2.6) / 8.6 (0.8) |
| MONet (Burgess et al., 2019) | 3.1 (1.6) / 7.0 (2.6) / 9.8 (3.6) / 1.2 (0.8) | 24.8 (1.6) / 24.6 (1.6) / 31.0 (1.6) / 40.7 (1.8) | 11.8 (2.0) / 12.5 (1.1) / 16.1 (0.9) / 21.9 (1.7) |
| IODINE (Greff et al., 2019) | 1.8 (0.2) / 3.9 (1.3) / 6.2 (2.0) / 7.3 (1.9) | 10.1 (2.9) / 13.7 (2.7) / 18.6 (4.2) / 24.4 (3.8) | 4.0 (1.2) / 6.3 (1.2) / 9.9 (1.8) / 10.8 (2.0) |
| Slot-Attention (Locatello et al., 2020) | 9.2 (0.4) / 13.5 (0.9) / 20.0 (1.3) / 26.2 (6.8) | 5.7 (0.3) / 9.0 (1.5) / 12.4 (2.5) / 18.3 (2.7) | 0.8 (0.3) / 3.5 (1.2) / 5.3 (1.7) / 7.3 (2.2) |
| I-SA (Chang et al., 2022) | 31.5 (15.2) / 25.6 (9.0) / 38.1 (12.5) / 40.2 (11.9) | 21.4 (6.8) / 23.4 (1.5) / 29.1 (7.8) / 34.5 (7.0) | 12.8 (4.8) / 13.7 (4.5) / 20.4 (6.0) / 24.6 (7.3) |
| Ours (transformer) | **48.0 (1.8) / 34.8 (1.3) / 50.8 (1.1) / 53.6 (0.7)** | **28.5 (2.4) / 26.4 (2.0) / 37.3 (2.0) / 42.4 (1.9)** | **17.8 (0.6) / 17.6 (0.6) / 25.3 (0.6) / 30.6 (0.9)** |

Table 5: Unsupervised foreground extraction results on CUB200 Birds (Birds), Stanford Dogs (Dogs), Stanford Cars (Cars), and Caltech Flowers (Flowers). We visualize the best results in bold.

| Model | Birds | | Dogs | | Cars | | Flowers | |
|---|---|---|---|---|---|---|---|---|
| | ↑ IoU | ↑ Dice | ↑ IoU | ↑ Dice | ↑ IoU | ↑ Dice | ↑ IoU | ↑ Dice |
| ReDO (Chen et al., 2019) | 46.5 | 60.2 | 55.7 | 70.3 | 52.5 | 68.6 | 76.4 | - |
| IODINE (Greff et al., 2019) | 30.9 | 44.6 | 54.4 | 67.0 | 51.7 | 67.3 | - | - |
| OneGAN (Benny & Wolf, 2020) | 55.5 | 69.2 | 71.0 | 81.7 | 71.2 | 82.6 | - | - |
| Slot-Attention (Locatello et al., 2020) | 35.6 | 51.5 | 39.6 | 55.3 | 41.3 | 58.3 | 30.8 | 45.9 |
| Voynov et al. (2020) | 68.3 | - | - | - | - | - | 54.0 | - |
| DRC (Yu et al., 2021) | 56.4 | 70.9 | 71.7 | 83.2 | 72.4 | 83.7 | - | - |
| Melas-Kyriazi et al. (2021) | 66.4 | - | - | - | - | - | 54.1 | - |
| SLATE (Singh et al., 2021) | 36.1 | 51.0 | 62.3 | 76.3 | 75.5 | 85.9 | 68.1 | 79.1 |
| I-SA (Chang et al., 2022) | 63.7 | 72.7 | 80.6 | 89.1 | 85.9 | 92.3 | 75.0 | 83.9 |
| Ours (mixture) | 25.1 | 39.2 | 36.8 | 53.6 | 69.1 | 81.5 | 36.1 | 51.6 |
| Ours (transformer) | **71.0** | **82.6** | **82.5** | **90.3** | **87.5** | **93.2** | **78.4** | **86.1** |

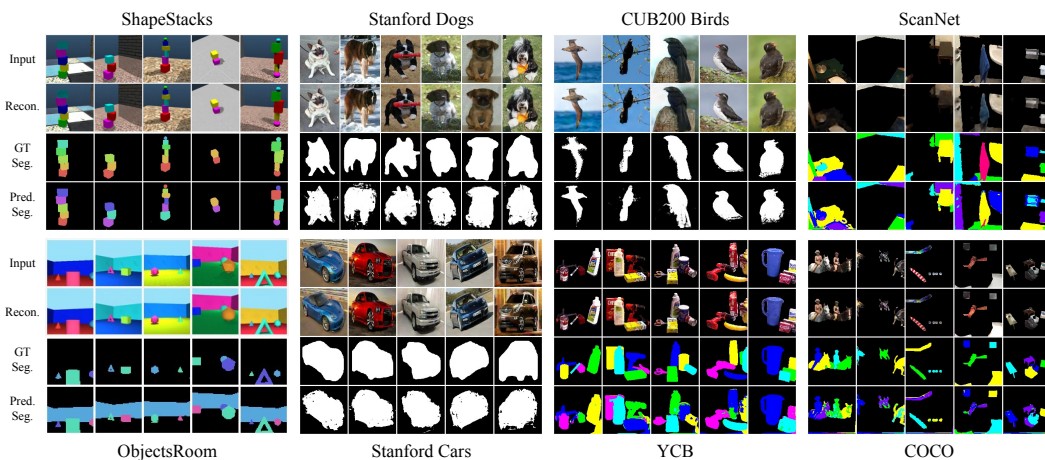

Figure 1: Visualization of our predicted segmentation and reconstruction results on synthetic and real images. We color the predicted mask that has a maximum intersection with the ground-truth background in black.

unsupervised foreground extraction, we follow Yu et al. (2021) and report the best model performance on all datasets. During the evaluation, we select the slot's mask prediction that has a maximum intersection with the ground-truth foreground mask as our predicted foreground. For unsupervised multi-object segmentation, we follow Yang & Yang (2022) and report the models' performance on all datasets across trials with different random seeds.

**Results** We show quantitative experimental results in Tab. 5 and Tab. 4. We also visualize qualitative results in Fig. 1. For multi-object segmentation, as shown in Tab. 4, our model outperforms existing object-centric learning baselines by a large margin, especially on the YCB dataset where the segmented objects have clear semantic meanings. For foreground extraction, as shown in Tab. 5, our method significantly outperforms all existing baselines on the task of foreground extraction, achieving new state-of-the-art on all datasets. We recognize the dis-

Table 6: Unsupervised segmentation results on Birds (mIoU↑). *Contrastive learning methods are pre-trained on ImageNet and segment with K-means clustering.

| Model | Birds |
|---|---|
| MoCo v2 (Chen et al., 2020) | 63.5 |
| BYOL (Grill et al., 2020) | 56.1 |
| R2O (Gokul et al., 2022) | **71.2** |
| ours (BO-QSA+transformer) | **71.0** |

Table 7: Ablative experiments on slot initialization and optimization methods. We visualize the best results in bold and underline the second-best results. (*Note that SA represents Slot-Attention with our encoder-decoder design and is different from the original one reported in Tab. 5.)

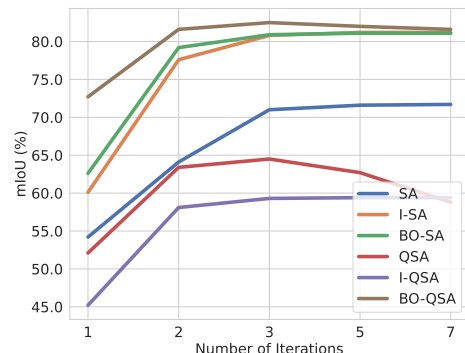

Figure 2: Effects of iterative updates in testing.

| Method | Dogs | | ShapeStacks | |
|---|---|---|---|---|
| | ↑ IoU | ↑ Dice | ↑ ARI-FG(%) | ↑ MSC-FG(%) |
| SA* | 71.0 | 81.9 | 86.7 | 84.8 |
| I-SA | 80.8 | 89.2 | 88.3 | 76.8 |
| BO-SA | 80.9 | 89.3 | 87.7 | 66.6 |
| QSA | 64.5 | 72.9 | 88.1 | 76.1 |
| I-QSA | 59.3 | 77.6 | 84.6 | 81.8 |
| BO-QSA (ours) | 82.5 | 90.3 | 92.9 | 89.2 |

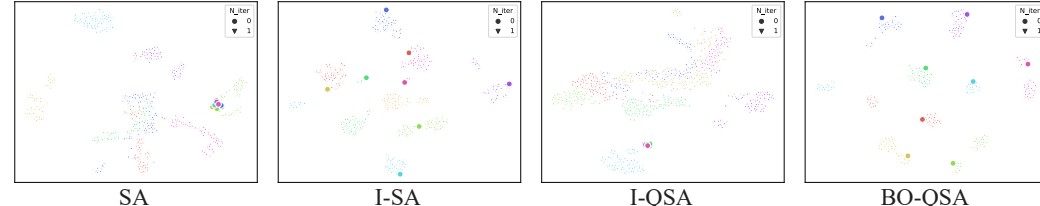

| SA | I-SA | I-QSA | BO-QSA |

Figure 3: Visualization of learned slot initializations and post-iteration slots after the first iteration of Slot-Attention on ShapeStacks (we use dots for initialization vectors and inverse triangles for post-iteration slots). We show our BO-QSA provides the best overall separation as well as correspondence between initialization vectors and post-iteration slots. For I-SA, there exist mismatches between initialization vectors and post-iteration slots (yellow and red). The same optimization method is also not effective for I-QSA, leading to mixing post-iteration slots similar to SA for slot initializations (best viewed in color and with zoom-in).

crepancy of mixture-based decoders in both Slot-Attention and our mixture-based design in modeling real-world images, reflecting similar discoveries from recent works (Singh et al., 2021) that mixture-based decoder struggles in modeling real-world images. On the other hand, our transformer-based model shows significant improvements over the vanilla version. Notably, our method outperforms a broad range of models, including GAN-based generative models (*i.e.* OneGAN, Voynov et al. (2020)), and large-scale pre-trained contrastive methods (*i.e.* MoCo-v2, BYOL, R2O). As shown in Tab. 6, our method achieves comparable results with state-of-the-art self-supervised contrastive learning methods without large-scale pre-training and data augmentation. This result sheds light on the potential of object-centric learning as a pre-training task for learning general visual representations.

## 5.3 ABLATIVE STUDIES

**Experimental Setup**    We perform ablative studies over our designs by comparing them with different design variants on ShapeStacks and Stanford Dogs. For slot initialization, we consider (1) the original Slot-Attention module's sampling initialization (SA), and (2) initializing with learnable queries (QSA). For optimization, we consider (1) the original optimization in Slot-Attention (*i.e.* w/o detach or STE), (2) the I-SA optimization where gradients to slots in iterative updates are detached (*i.e.* w/ detach only), and (3) our optimization where we both detach the gradients into iterative refinement, and pass gradient to the initialization queries with STE (*i.e.* w/ detach and STE). For simplicity, we term these variants with prefixes (I-) for I-SA and (BO-) for our full method. We run all ablations on each dataset with the same encoder-decoder architecture.

**Results**    We show experimental results in Tab. 7 and Fig. 2. First, from Tab. 7, we observe that BO-QSA significantly outperforms other variants. For sample-based slot initializations, our method shows a similar effect compared with I-SA on improving Slot-Attention learning. For query-based slot initializations, we validate the difficulty in training query-based Slot-Attention with its inferior performance. We further show the ineffectiveness of I-SA for query-based Slot-Attention. The experiments on query-based Slot-Attention prove that both of our design choices are necessary and effective for superior performance. To study the effect of learned queries, we visualize in Fig. 2 where we set different numbers of iterative updates of Slot-Attention during inference on the Stanford

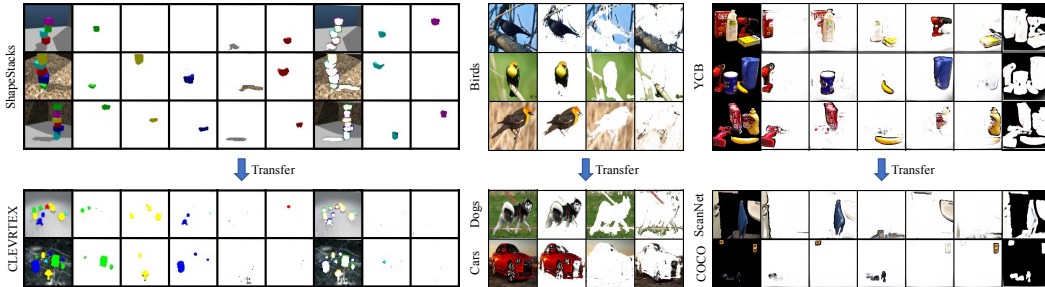

Figure 4: Visualization of learned concepts and attention maps in zero-shot transfer. At the top, we visualize the per-slot reconstruction of our model trained on ShapeStacks (left), Birds (middle), and YCB (right). At the bottom, we show that our learned slot initialization queries bind to the same concepts in zero-shot transfer experiments (*i.e.* color in ShapeStacks to CLEVRTEX, contours in Birds to Dogs and Cars, and spatial positions in YCB to ScanNet and COCO) by visualizing attention maps of slot initialization queries over input images. *Note that for the ShapeStacks experiment(left), we alternate object colors in CLEVRTEX with seen colors for better qualitative evaluations, and we do not perform such operations for quantitative evaluations.

Dogs dataset. We can see that our BO-QSA significantly outperforms other variants with only one iteration. This indicates that our query-based design can help ease training difficulties. In Fig. 3, we further visualize the learned initializations and post-iteration slots in the same feature space using t-SNE (Van der Maaten & Hinton, 2008). Our initializers provide a more separable space when differentiating image features, which validates the desired model behaviors mentioned in Sec. 3.3.

### 5.4 ADDITIONAL ANALYSES

In this section, we provide additional analyses on the potential of our BO-QSA as a concept binder for generalizing to new examples. First, we qualitatively visualize our learned content for each slot (without additional clustering) in ShapeStacks, Birds, and YCB in Fig. 4. We observe high similarity within the learned content of each slot, indicating similar concepts learned by specific slots. This shows the potential of the slots in our BO-QSA for binding specific concepts on object properties (*e.g.* colors, contours, and spatial positions). Although we can not control which concepts to learn, these results are important indicators that our learned initialization queries could potentially be generalizable concept probes. We further

provide quantitative evaluations where we use models trained on dataset X for zero-shot inference on dataset Y. We term this transfer as (X→Y). As shown in Tab. 8, when adapting models trained on YCB to zero-shot inference on ScanNet and COCO, our method outperform I-SA and also the majority of fine-tuned methods shown in Tab. 4. Due to the page limit, we show in Appendix B.1 that this superior transfer capability is general across datasets when compared to Slot-Attention variants.

Table 8: Zero-shot transfer results of unsupervised multi-object segmentation on real images.

| Model | YCB → ScanNet (AP / PQ / Pre / Rec) | YCB → COCO (AP / PQ / Pre / Rec) |
|---|---|---|
| SA | 1.37 / 4.90 / 11.27 / 6.35 | 1.20 / 4.97 / 10.48 / 6.73 |
| I-SA | 21.62 / 21.81 / 32.32/ 34.19 | 18.39 / 18.47 / 27.23 / 30.38 |
| BO-QSA (ours) | **28.24 / 25.93 / 36.68 / 39.62** | **24.23 / 21.65 / 30.20 / 35.79** |

### 6 CONCLUSIONS

We introduce BO-QSA for unsupervised object-centric representation learning. We initialize Slot-Attention with learnable queries, and combine bi-level optimization and straight-through gradient estimators to ease the difficulty in query-based Slot-Attention learning. With simple code adjustments on Slot-Attention, we obtain state-of-the-art model for unsupervised object segmentation in both synthetic and natural image domains, outperforming previous baselines by a large margin. More importantly, our learned model exhibits concept-binding effects where visual concepts are attached to specific slot queries. With a fixed number of initialized slots, our model is limited to handling a fixed maximum number of objects in the inputs. However, our queries could be learned to bind object attributes, which leads to meaningful segmentation of images by grouping similar properties (*e.g.* color, position, *etc.*). As a future direction, this connects our method with weakly-supervised contrastive learning methods that learn grounded visual representations with language.

## ACKNOWLEDGEMENT

We gratefully thank all colleagues from BIGAI for fruitful discussions. We would also like to thank the anonymous reviewers for their constructive feedback. This work reported herein was supported by National Key R&D Program of China (2021ZD0150200).

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

# A  MODEL ARCHITECTURE AND DESIGN

## A.1  DESIGN OF DECODERS

In this section, we follow the notations used in Sec. 2.1 and describe two common approaches, mixture-based and transformer-based, for decoding images from the learned slot representations.

**Mixture-based Decoder**  The mixture-based decoder (Watters et al., 2019) decodes each slot $\hat{\mathbf{s}}_i$ into an object image $\mathbf{x}_i$ and mask $\mathbf{m}_i$ with decoding functions $g_{\phi^{\text{dec}}}^{\text{img}}$ and $g_{\phi^{\text{dec}}}^{\text{mask}}$, which are implemented using CNNs. The decoded images and masks are calculated by:

$$\hat{\boldsymbol{I}}_i = g_{\phi^{\text{dec}}}^{\text{img}}(\hat{\mathbf{s}}_i), \quad \mathbf{m}_i = \frac{\exp\ g_{\phi^{\text{dec}}}^{\text{mask}}(\hat{\mathbf{s}}_i)}{\sum_{j=1}^{K} \exp\ g_{\phi^{\text{dec}}}^{\text{mask}}(\hat{\mathbf{s}}_j)}, \quad \hat{\boldsymbol{I}} = \sum_{i=1}^{K} \mathbf{m}_i \cdot \hat{\boldsymbol{I}}_i.$$

During training, a reconstruction objective is employed for supervising model learning. Despite its wide usage, mixture-based decoders showed limited capability at handling natural scenes with high visual complexity (Singh et al., 2021).

**Autoregressive Transformer Decoder**  Recently, Singh et al. (2021; 2022) reveal the limitations of mixture decoder and leverage transformers and dVAEs (Van Den Oord et al., 2017; Ramesh et al., 2021) for decoding slot-based object-centric representations. To obtain decoded images $\hat{\boldsymbol{I}}$, they learn a separate dVAE for first encoding $\boldsymbol{I}$ into a sequence of $L$ tokens $\mathbf{z} = \{\mathbf{z}_1, \cdots, \mathbf{z}_L\}$ with dVAE encoder $f_{\phi^{\text{enc}}}^{\text{dVAE}}$. Next, they use a transformer decoder $g_{\phi^{\text{dec}}}^{\text{transformer}}$ to auto-regressively predict image tokens with learned slot representation $\hat{\mathbf{s}}$:

$$\mathbf{o}_l = g_{\phi^{\text{dec}}}^{\text{transformer}}(\hat{\mathbf{s}}; \mathbf{z}_{<l}) \quad \text{where} \quad \mathbf{z} = f_{\phi^{\text{enc}}}^{\text{dVAE}}(\boldsymbol{I}).$$

To train the entire model, we have the reconstruction objective supervising the learning of $\mathbf{z}$ with dVAE decoder $g_{\phi^{\text{dec}}}^{\text{dVAE}}$. Next, the objective for object-centric learning relies on the correct prediction from the auto-regressive transformer for predicting correct tokens:

$$\mathcal{L} = \mathcal{L}_{\text{dVAE}} + \mathcal{L}_{\text{CE}} \quad \text{where} \quad \mathcal{L}_{\text{dVAE}} = ||g_{\phi^{\text{dec}}}^{\text{dVAE}}(\mathbf{z}) - \boldsymbol{I}||_2^2, \ \mathcal{L}_{\text{CE}} = \sum_{l=1}^{L} \text{CrossEntropy}(\mathbf{z}_l, \mathbf{o}_l)$$

Under this setting, the model does not predict additional masks and relies on the attention $\boldsymbol{A}$ within the Slot-Attention module for obtaining slot-specific object masks. Although such models can achieve competitive results on real-world synthetic datasets, as our experiments suggest, they can be inferior to mixture-based decoders on segmentation in synthetic datasets. We suspect that this originates from the low resolution when discretizing images into tokens.

## A.2  BI-LEVEL OPTIMIZATION AND META-LEARNING

Recall the bi-level optimization problem we introduced in Sec. 2.2.

$$\min_{\theta, \phi} f(\theta, \phi) \quad s.t. \quad \theta \in \arg\min_{\theta'} g(\theta', \phi), \tag{6}$$

where we call $f(\theta, \phi)$ the outer objective function and $g(\theta, \phi)$ the inner objective function. To jointly optimize both objectives w.r.t. parameters $\theta$ and $\phi$, a straightforward approach to solving Eq. (6) is to represent the inner solution of $\theta$ as a function of $\phi$, *i.e.*, $\theta^*(\phi) = \arg\min_{\theta'} g(\theta', \phi)$. Then we can optimize the outer objective with gradient descent:

$$\nabla_{\phi} f(\theta^*(\phi), \phi) = \nabla_{\phi}\theta^*(\phi)\nabla_1 f(\theta^*(\phi), \phi) + \nabla_2 f(\theta^*(\phi), \phi),$$

However, the difficulty of this method lies in the calculation of $\nabla_{\phi}\theta^*(\phi)$ where we need to solve linear equation from implicit gradient theorem:

$$\nabla_{1,2} g(\theta^*(\phi), \phi)\nabla_{\phi}\theta^*(\phi) + \nabla_{2,2} g(\theta^*(\phi), \phi) = 0.$$

If $\nabla_{2,2} g(\theta^*, \phi)$ is invertible, we can solve for $\nabla_{\phi}\theta^*(\phi)$ and obtain the gradient update on $\phi$:

$$\phi_{k+1} = \phi_k - \xi \left( \nabla_2 f_k - (\nabla_{1,2} g_k)^{\top} (\nabla_{2,2} g_k)^{-1} \nabla_1 f_k \right)$$

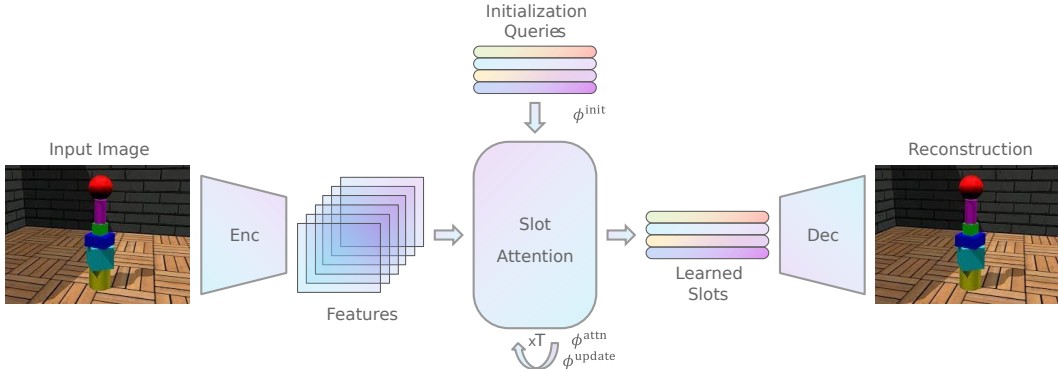

Figure 5: An illustrative visualization of our proposed BO-QSA slot-encoder. During the backward pass, BO-QSA uses STE to backpropagate gradients directly to $\phi^{\text{init}}$, $\phi^{\text{attn}}$, and $\phi^{\text{update}}$ without gradients into the iterative process.

where $\nabla_1 f_k = \nabla_2 f(\theta^*(\phi_k), \phi_k)$ and $\nabla_1 f_k = \nabla_1 f(\theta^*(\phi_k), \phi_k)$. Various methods have been proposed to approximate the solution (Pedregosa, 2016; Lorraine et al., 2020), and we refer the authors to Ye et al. (2022) for a thorough review of related methods.

Bi-level optimization is closely related to meta-learning. In meta-learning, we have meta-training tasks which comes in as $N$ different collections of datasets $\mathcal{D} = \{\mathcal{D}_i = \mathcal{D}_i^{\text{tr}} \cup \mathcal{D}_i^{\text{val}}\}_{i=1}^N$. The inner and outer objectives in Eq. (6) are substituted by averaging training and validation errors over multiple tasks (Franceschi et al., 2018):

$$\min_{\theta, \phi} f(\theta, \phi) = \sum_{i=1}^N \mathcal{L}_i(\theta_i, \phi, \mathcal{D}_i^{\text{val}}) \quad s.t. \quad \theta_i = \min_{\theta_i'} \sum_{i=1}^N \mathcal{L}_i(\theta_i', \phi; \mathcal{D}_i^{\text{tr}}), \tag{7}$$

where $\mathcal{L}_i$ represents task-dependent error on $\mathcal{D}_i$. The final goal of meta-learning aims at seeking the meta-parameter $\phi$ that is shared between tasks which later enables few-shot learning and fast adaptation. With its connections with bi-level optimization, the previously mentioned optimization methods are broadly adapted for solving meta-learning problems (Finn et al., 2017; Nichol & Schulman, 2018; Rajeswaran et al., 2019). From the meta-learning perspective, our attempt shares similar insights with first-order meta-learning methods (Finn et al., 2017; Nichol & Schulman, 2018), where we use the gradient at some task-specific optimal solution $\mathbf{s}_i^*$ of the inner optimization for optimizing slot initialization queries which are shared across datasets on the outer objective. This meta-learning perspective also indicates the potentials of our BO-QSA for fast adaptation and generalization.

## A.3 IMPLEMENTATION DETAILS

We provide a visualization of our designed slot-encoder in Fig. 5 and discuss the implementation details for different experimental settings in the following sections.

### A.3.1 SLOT INITIALIZATION

We initialize all models with the number of slots shown in Tab. 13. During training, we add a small perturbation to the queries by sampling from a zero-mean distribution with variance $\sigma$ as we found it empirically helpful for better performance. We perform annealing over $\sigma$ to gradually eliminate the effect of this random perturbation during training. We adopt the cosine annealing strategy such that $\sigma$ starts from 1 and gradually anneals to 0 after $N_\sigma$ training steps, where $N_\sigma$ is a hyperparameter that controls the annealing rate of $\sigma$. In our experiments, we use $N_\sigma = 0$ on Cars and Flowers and $N_\sigma = 30000$ on the rest of the datasets.

| Layer | Kernel Size | Stride | Padding | Channels | Activation |
|-------|-------------|--------|---------|----------|------------|
| Conv | 5x5 | 1(2) | 2 | 64 | ReLU |
| Conv | 5x5 | 1 | 2 | 64 | ReLU |
| Conv | 5x5 | 1 | 2 | 64 | ReLU |
| Conv | 5x5 | 1 | 2 | 64 | ReLU |

Table 9: Configuration of CNN encoder used in our model. The values in parentheses are adopted for CLEVRTex and ShapeStacks

| Layer | Kernel Size | Stride | Padding | Channels | Activation |
|-------|-------------|--------|---------|----------|------------|
| TransConv | 5x5 | 2 | 2 | 64 | ReLU |
| TransConv | 5x5 | 2 | 2 | 64 | ReLU |
| TransConv | 5x5 | 2 | 2 | 64 | ReLU |
| TransConv | 5x5 | 2(1) | 2 | 64 | ReLU |
| TransConv | 5x5 | 1 | 2 | 64 | ReLU |
| TransConv | 3x3 | 1 | 1 | 4 | None |

Table 10: Configuration of mixture decoder used in our model. The values in parentheses are adopted for ObjectsRoom

### A.3.2 BO-QSA with Mixture-Based Decoders

For mixture-based decoders, we use the same Slot-Attention architecture as in Locatello et al. (2020) with slots initialized by learnable queries. Given an input image, Slot-Attention uses a CNN encoder to extract image features. After adding positional embedding, these features are input into the Slot-Attention module slot updates. Finally, these slots are decoded by the mixture decoder to reconstruct the input image. We provide the details of our image encoder in Tab. 9. For the mixture-based decoder, we use six transposed convolutional layers with ReLU activations following Locatello et al. (2020). We visualize the details of our mixture-based decoder design in Tab. 10. We train our model for 250k steps with a batch size of 128 and describe all training configurations and hyperparameter selection Tab. 11.

| Batch Size | LR | Slot Dim | MLP Hidden Dim |
|------------|-----|----------|----------------|
| 128 | 4e-4 | 64 | 128 |

| Warmup Steps | Decay Steps | Max Steps | Sigma Down Steps |
|--------------|-------------|-----------|------------------|
| 5k | 50k | 250k | 30k |

Table 11: Training configuration for mixture-based model

### A.3.3 BO-QSA with Transformer-Based Decoder

For transformer-based decoders, we adopt the transformer architecture proposed by SLATE (Singh et al., 2021). For the transformer-based BO-QSA, unlike SLATE, we use the same CNN as in mixture-based BO-QSA (instead of the dVAE encoder) to extract features from the image as input to the Slot-Attention module as we find such changes help solve the problem on coarse object boundary prediction mentioned in Sec. 5.1. Next, we use the same overall architecture of dVAE as mentioned in SLATE Singh et al. (2021). However, we change the kernel size of the dVAE encoder from 1 to 3 since we find that such changes can help increase model performance when decomposing scenes. We train our model for 250k steps with a batch size of 128, and all the training configuration in our experiments is described in Tab. 12.

### A.3.4 Baselines

The reproduction of Slot-Attention and SLATE follows the architecture and hyperparameter selection mentioned in their paper. Similar to our models, we train all baseline models with 250K steps on all datasets. For SLATE, we use the input image size of 96 on the ShapeStacks dataset as we find that the image size of 128 will cause all objects to be divided into the same slot, resulting in low

| | | | |
|---|---|---|---|
| Training | batch size | | 128 |
| | warmup steps | | 10000 |
| | learning rate | | 1e-4 |
| | max steps | | 250k |
| dVAE | vocabulary size | | 1024 |
| | Gumbel-Softmax annealing range | | 1.0 to 0.1 |
| | Gumbel-Softmax annealing steps | | 30000 |
| | lr-dVAE(no warmup) | | 3e-4 |
| Transformer Decoder | layers | | 4 |
| | heads | | 4 |
| | dropout | | 0.1 |
| | hidden dimension | | 256 |
| Slot Attention Module | slot dimension | | 256 |
| | iterations | | 3 |
| | $\sigma$ annealing steps | | 30000(0) |

Table 12: Training configuration for transformer-based model. The values in parentheses are adopted for Cars and Flowers dataset

ARI and MSC. For a fair comparison with numbers reported in SLATE's paper, we report the MSE of models by first computing per-pixel errors and then multiplying it by the total number of pixels. For CLEVRTEX, we follow the same experimental setting of (BO-QSA+mixture) for ShapeStacks and set the number of slots to 11. For YCB, ScanNet, and COCO, we follow the same experimental setting of (BO-QSA+transformer) for birds and set the number of slots to 6.

| | Model | Shapestacks | ObjectsRoom | Birds | Dogs | Flowers | Cars |
|---|---|---|---|---|---|---|---|
| # of slots | Slot-Attention | 8 | 5 | 3 | 2 | 2 | 2 |
| | SLATE | 12 | 6 | 3 | 2 | 2 | 2 |
| | BO-QSA +Mixture | 8 | 5 | 3 | 2 | 2 | 2 |
| | BO-QSA +Transformer | 12 | 6 | 3 | 2 | 2 | 2 |
| | Image Size | 128 | 64 | 128 | 128 | 128 | 128 |

Table 13: The number of slots and image size used for each dataset

# B ADDITIONAL EXPERIMENTS

## B.1 ZERO-SHOT TRANSFER

In this section, we continue the discussion in Sec. 5.4 and provide additional zero-shot transfer results. Similarly, we use the notation $(X \to Y)$ to denote the zero-shot adaptation of models trained unsupervisedly on dataset $X$ to new datasets $Y$.

For unsupervised multi-object segmentation, we report transfer results from ScanNet and COCO to all other real-image multi-object segmentation datasets in addition to the results on YCB (mentioned in Sec. 5.4). As shown in Tab. 14, our model shows consistent improvement over Slot-Attention and I-SA during zero-shot transfer.

Table 14: Zero-shot transfer results of unsupervised multi-object segmentation on real images.

| Model | ScanNet → YCB (AP / PQ / Pre / Rec) ↑ | ScanNet → COCO (AP / PQ / Pre / Rec) ↑ | COCO → YCB (AP / PQ / Pre / Rec) ↑ | COCO → ScanNet (AP / PQ / Pre / Rec) ↑ |
|---|---|---|---|---|
| SA | 19.63 / 19.24 / 28.56 / 31.43 | 12.84 / 14.86 / 22.06 / 26.74 | 26.53 / 23.05 / 35.96 / 38.12 | 20.99 / 22.08 / 32.14 / 36.53 |
| I-SA | 18.66 / 18.56 / 28.97 / 30.82 | 11.83 / 14.14 / 20.70 / 25.42 | 26.72 / 22.90 / 35.89 / 37.98 | 19.34 / 20.00 / 29.44 / 33.18 |
| BO-QSA (ours) | **21.85 / 19.96 / 31.51 / 33.45** | **13.95 / 16.04 / 23.35 / 28.49** | **31.21 / 25.44 / 38.90 / 41.35** | **24.21 / 23.59 / 34.07 / 38.49** |

For unsupervised foreground extraction, we report transfer results from Stanford Dogs and CUB200 Birds to all other real-image foreground extraction datasets. As we can see from Tab. 15, our model

achieves the overall best results compared with other powerful Slot-Attention variants (models that achieve best or second-best results in our ablation studies as in Tab. 7) except for (Birds→Cars). However, our optimization method still helps improve zero-shot transfer for randomly initialized Slot-Attention.

Table 15: Zero-shot transfer results on unsupervised foreground extraction (mIoU ↑).

| Model | Dogs → Cars | Dogs → Flowers | Dogs → Birds | Birds → Dogs | Birds → Cars | Birds → Flowers |
|---|---|---|---|---|---|---|
| SA | 57.96 | 57.96 | 45.06 | 74.68 | 58.79 | 62.02 |
| I-SA | 58.05 | 58.06 | 48.88 | 71.16 | 69.90 | 68.67 |
| BO-SA | 58.10 | 58.10 | 47.96 | 71.81 | **70.75** | 67.95 |
| BO-QSA (ours) | **75.50** | **63.43** | **52.49** | **76.66** | 66.74 | **70.74** |

## B.2 ANALYSIS NUMBER OF SLOT-ATTENTION ITERATIONS

As described in Sec. 3.2, we study whether a fixed point $s^*$ could be reached by a fixed number of iterations during training. Since we hypothesized that the low performance of I-QSA in Sec. 5.3 originated from the insufficient number of starting points for fixed-point approximation, we conduct experiments on increasing the number of Slot-Attention iterations during training for I-QSA on the Dog dataset. As shown in Tab. 16, increasing the number of Slot-Attention iterations during training for I-QSA significantly improves its performance. However, we found that adding more iterations after a threshold (*i.e.* 7 in this case) does not further improve the overall performance. This verifies the need for learning slot initialization vectors for better approximating the fixed point solution of the inner soft-clustering objective in Slot-Attention.

Table 16: Increasing the number of iterations during training for I-QSA.

| Model | # of Training Iterations | Dogs | | | |
|---|---|---|---|---|---|
| | | ↑ IoU | Gain | ↑ Dice | Gain |
| I-QSA | 3 | 59.3 | - | 77.6 | - |
| I-QSA | 7 | 80.5 | +35.8% | 88.9 | +14.6% |
| Ours | 3 | **82.5** | - | **90.3** | - |

## B.3 DESIGN CHOICES ON SLOT INITIALIZATION

As described in Sec. 3.3, our method is connected with recent works on dVAE. However, we do not require the initialization queries to maintain information about the post-iteration slots $\hat{s}$ as we found such constraints lead to the learning of the mean representation of datasets which forbids disentanglement and concept binding. In this section, we provide experimental results to verify this argument. Specifically, we consider three different ways to update slot initialization queries in addition to our proposed method: 1) using the running mean of the post-iteration slots as initialization queries (RunningMean), 2) running K-Means clustering on post-iteration slots and updating the initialization queries using re-clustered centers by Hungarian matching (KMeans), 3) adding consistency loss between initialization queries and post-iteration slots as done in VQ-VAE (VQ-constraint). For (1) and (2), we empirically found such designs to be suffering from frequent updates and therefore use momentum updates to stabilize their training. We term these variants with the suffix (-M).

Table 17: Comparison between update methods for slot-initialization queries.

| Metrics | RunningMean | RunningMean-M | KMeans | KMeans-M | VQ-constraint | Ours |
|---|---|---|---|---|---|---|
| ARI-FG (ShapeStacks) | 7.5 | 51.4 | 21.0 | 70.6 | 88.6 | **92.9** |
| MSC-FG (ShapeStacks) | 3.7 | 15.4 | 4.2 | 60.4 | 85.3 | **89.2** |

As shown in Tab. 17, our model achieves the best overall performance compared to other initialization methods. Specifically, we found that using the running mean of post-iteration slots or K-Means cluster centers re-clustered from post-iteration slots to be harmful to model performance. We attribute this

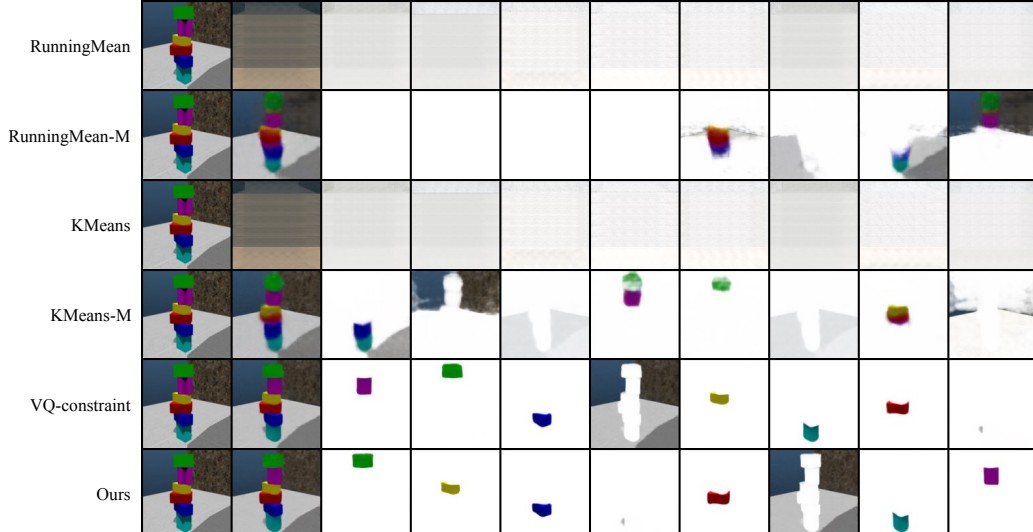

Figure 6: Visualizations per-slot reconstruction for different update methods. We show that RunningMean and KMeans suffer at decomposing the image, even with momentum updates. For VQ-constraint, though the model variant achieves a similar but slightly inferior effect on segmentation, they can not preserve the same filtered property for each slot across images.

effect to the learning of the mean-representation of datasets. This is further proved in experiments with VQ-VAE loss on consistency between slot initializations and post-iteration slots (*i.e.* $||\text{sg}(\hat{s}) - s_0||^2$), where the VQ-constraint variant showed inferior performance. We also found that the weight of this additional loss needs to be carefully tuned for the model to decompose objects. Empirically, most configurations of this hyperparameter will lead to bad reconstructions except for certain small weights (*e.g.* 0.01 reported here). Above all, we believe these experimental results verify the effectiveness of our design choices on initialization query learning. We provide additional visualizations on the learned contents of slots for each update method in Fig. 6.

## B.4 EXPERIMENTS ON ADDITIONAL DATASETS

In addition to datasets considered in Sec. 5, we conduct experiments on other synthetic datasets and visualize qualitative results. More specifically, we test our model on PTR (Hong et al., 2021). PTR is a synthetic dataset of 3D objects from PartNet with rendering variations. We run our BO-QSA with the same configuration mentioned in Appendix A.3 previously. We compare our method with the vanilla Slot-Attention module on multi-object segmentation. We report ARI-FG and MSC-FG scores of our model compared with the vanilla Slot-Attention on the PTR validation set.

Table 18: Multi-object segmentation results on PTR. We visualize the best results in bold.

| Model | PTR | |
|---|---|---|
| | ARI-FG ↑ | MSC-FG ↑ |
| Slot-Attention | 0.72 | 0.21 |
| ours (BO-QSA+mixture) | **0.75** | **0.61** |

As we can see from Tab. 18, our model achieves similar performance compared with Slot-Attention on ARI-FG and significantly outperforms it on MSC-FG. We attribute this result to the capability of precisely segmenting objects. As ARI-FG applies masks to each slot prediction for calculating results, it does not require models to precisely segment the object from the background. However, MSC-FG uses a mIoU-like measure that requires the model to precisely predict the object boundaries. This indicates that our model is better at precisely segmenting objects without noise. Similarly, we observe the binding of certain slots to scene backgrounds, but with more complex concepts, the binding of slots to concepts is not as straightforward as in ShapeStacks and CUB200 Birds.

To further investigate the effectiveness and generality of our method, we adapt BO-QSA to the recent 3D object-centric learning model, uORF (Yu et al., 2022), and test it on 3D datasets including CLEVR-567, Room-Chair, and Room-Diverse. uORF can decompose complex 3D scenes from a single image by combining NeRF (Mildenhall et al., 2021) with Slot-Attention. We only modify the initialization and optimization method of the Slot-Attention module in uORF, leaving all other hyperparameters unchanged. As we can see from Tab. 19, with our method, the uORF model that trained with 600 epochs can achieve a similar or even superior result compared to the original model trained with 1200 epochs. Additionally, when the dataset complexity increases (*e.g.*, in Room-Diverse), our method demonstrates significant improvement. Please refer to uORF (Yu et al., 2022) for more details about the model, datasets, and evaluation metrics.

Table 19: 3D-object segmentation results on CLEVR-567, Room-Chair, and Room-Diverse. We visualize the best results in bold and underline the second-best results. *indicates reimplemented results.

| Dataset | Model | Train-epoch | NV-ARI↑ | ARI↑ | ARI-FG↑ | LPIPS↓ | SSIM↑ | PSNR↑ |
|---|---|---|---|---|---|---|---|---|
| CLEVR-567 | uORF | 600* | 66.8 | 73.8 | 81.0 | 0.1249 | 0.8763 | 27.84 |
| | | 1200 | **84.4** | **87.4** | 85.3 | 0.0869 | 0.8985 | 29.32 |
| | uORF+BO-QSA | 600 | 74.5 | 82.9 | 89.1 | 0.0783 | 0.9153 | 30.07 |
| | | 1200 | 77.7 | 86.9 | **89.5** | **0.0711** | **0.9223** | **30.64** |
| Room-Chair | uORF | 600* | 37.9 | 39.4 | 18.8 | 0.2932 | 0.7734 | 25.08 |
| | | 1200 | 77.9 | 80.3 | 91.8 | 0.0845 | 0.8762 | 29.66 |
| | uORF+BO-QSA | 600 | 76.9 | 79.8 | **94.6** | 0.0821 | 0.8850 | 30.13 |
| | | 1200 | **80.5** | **83.2** | 93.8 | **0.0733** | **0.8938** | **30.61** |
| Room-Diverse | uORF | 120* | 51.2 | 60.1 | 62.0 | 0.2139 | 0.6905 | 25.21 |
| | | 240 | 56.6 | 68.5 | 66.7 | 0.1820 | 0.7146 | 25.92 |
| | uORF+BO-QSA | 120 | 60.4 | 70.0 | 75.1 | 0.1657 | 0.7137 | 26.38 |
| | | 240 | **63.0** | **72.8** | **76.6** | **0.1533** | **0.7378** | **26.85** |

## C  LIMITATIONS AND FUTURE WORK

We discuss all limitations of our work found in the experiments. First, we observed a strong correlation between the powerfulness of encoder-decoder architectures and model performance. However, in contrast to supervised learning, more powerful encoders/decoders do not guarantee superior performance. Gaining insights from how contrastive learning methods have shown the effect of concept emergence with large-scale pretraining, we can also incorporate such representations learned by self-supervised learning into object-centric learning to unite the best of both worlds. Second, our work is primarily limited by the fixed number of slot initialization vectors. In contrast to the vanilla Slot-Attention that could generalize to a new number of objects, our model can not easily generalize to scenarios with new concepts since our model learns a fixed set of separating spaces that best disentangle different parts of the image. This problem is also frequently met in semantic segmentation and object classification, where we can only use existing concepts to interpret novel objects/semantic entities. Although solutions to this close-vocabulary problem have been proposed in supervised classification and segmentation, we leave the exploration of this problem in object-centric learning to future work. Finally, the current learned slot initialization vectors do not explicitly bind towards concepts and need to be mined by humans. We believe this is an important next step in our current work to combine unsupervised object-centric learning with semantic alignments from language for concept grounding. This opens future research directions on learning finer-level organization of object concepts under more complex scenarios (*e.g.* hierarchical grouping) with weak supervision of correspondence.

# D    ADDITIONAL VISUALIZATIONS

We provide more qualitative results of our model on different datasets in the following pages.

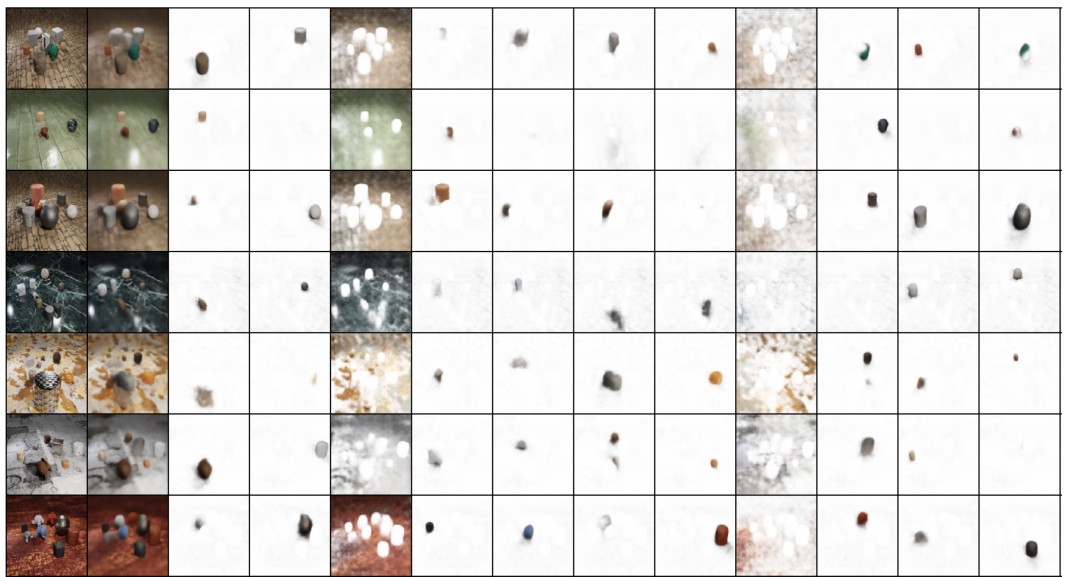

Figure 7: Unsupervised Multi-Object Segmentation on CLEVRTEX.

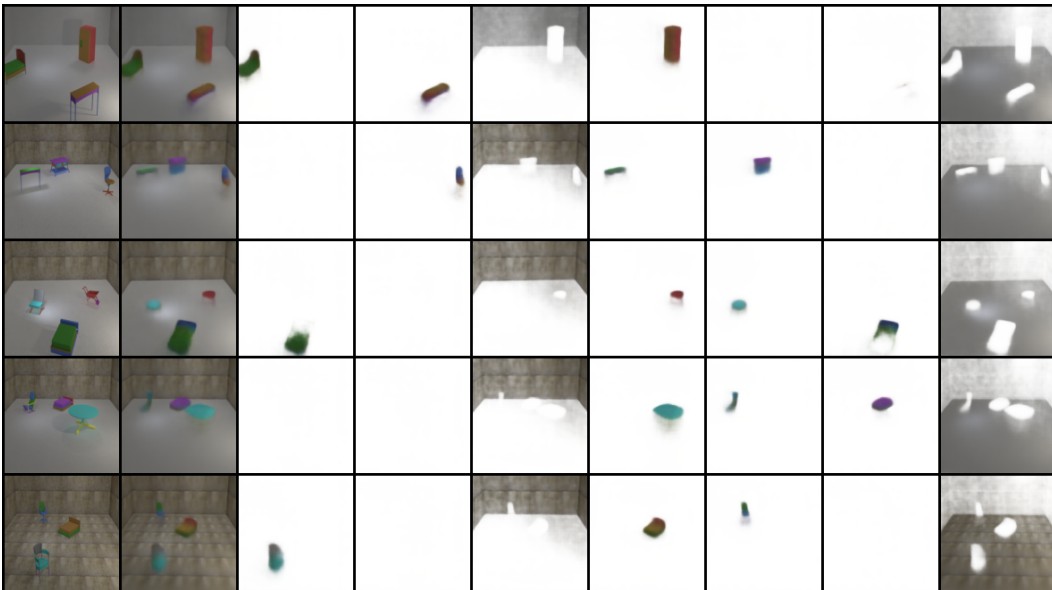

Figure 8: Unsupervised Multi-Object Segmentation on PTR.

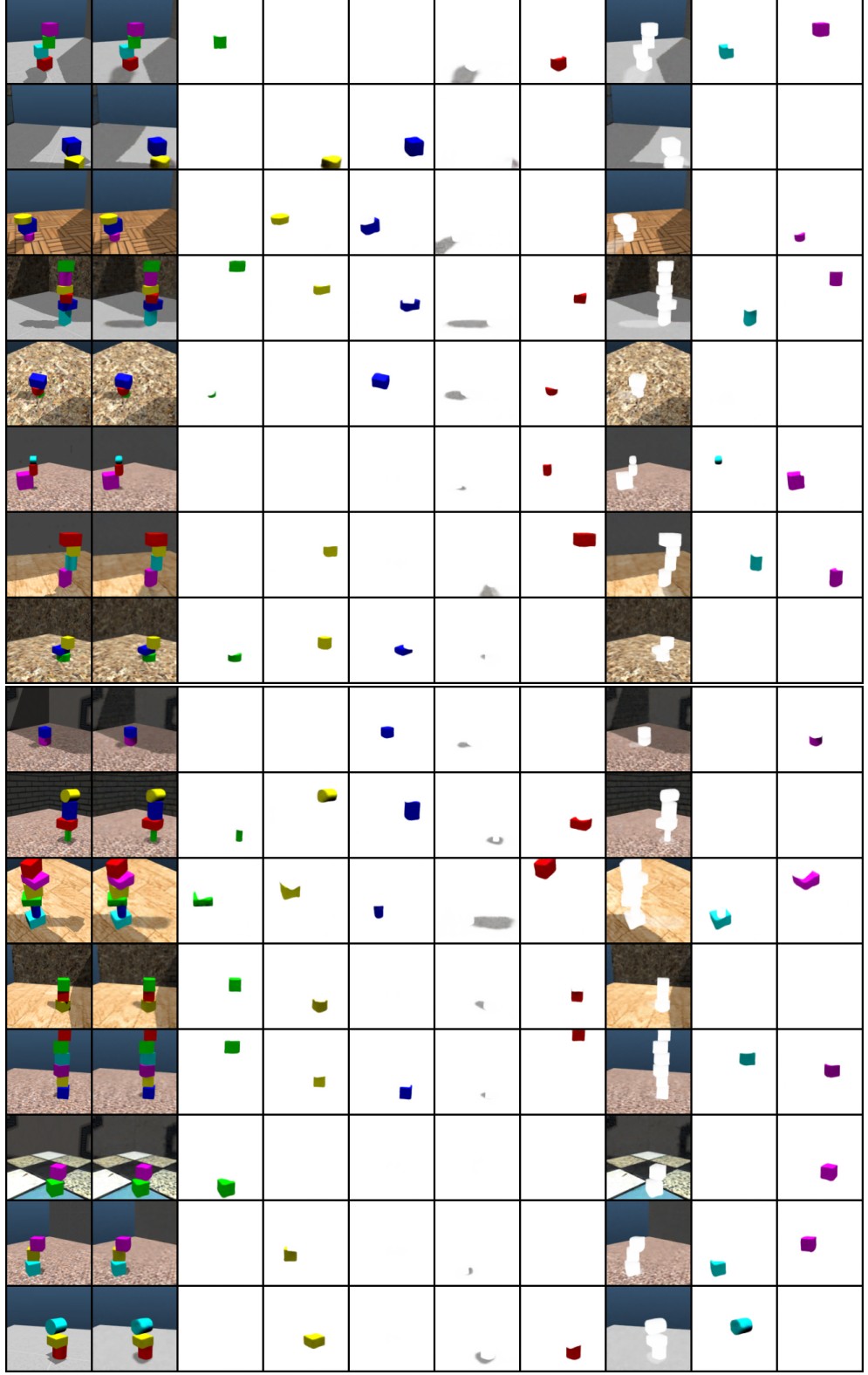

Figure 9: Unsupervised Multi-Object Segmentation on ShapeStacks.

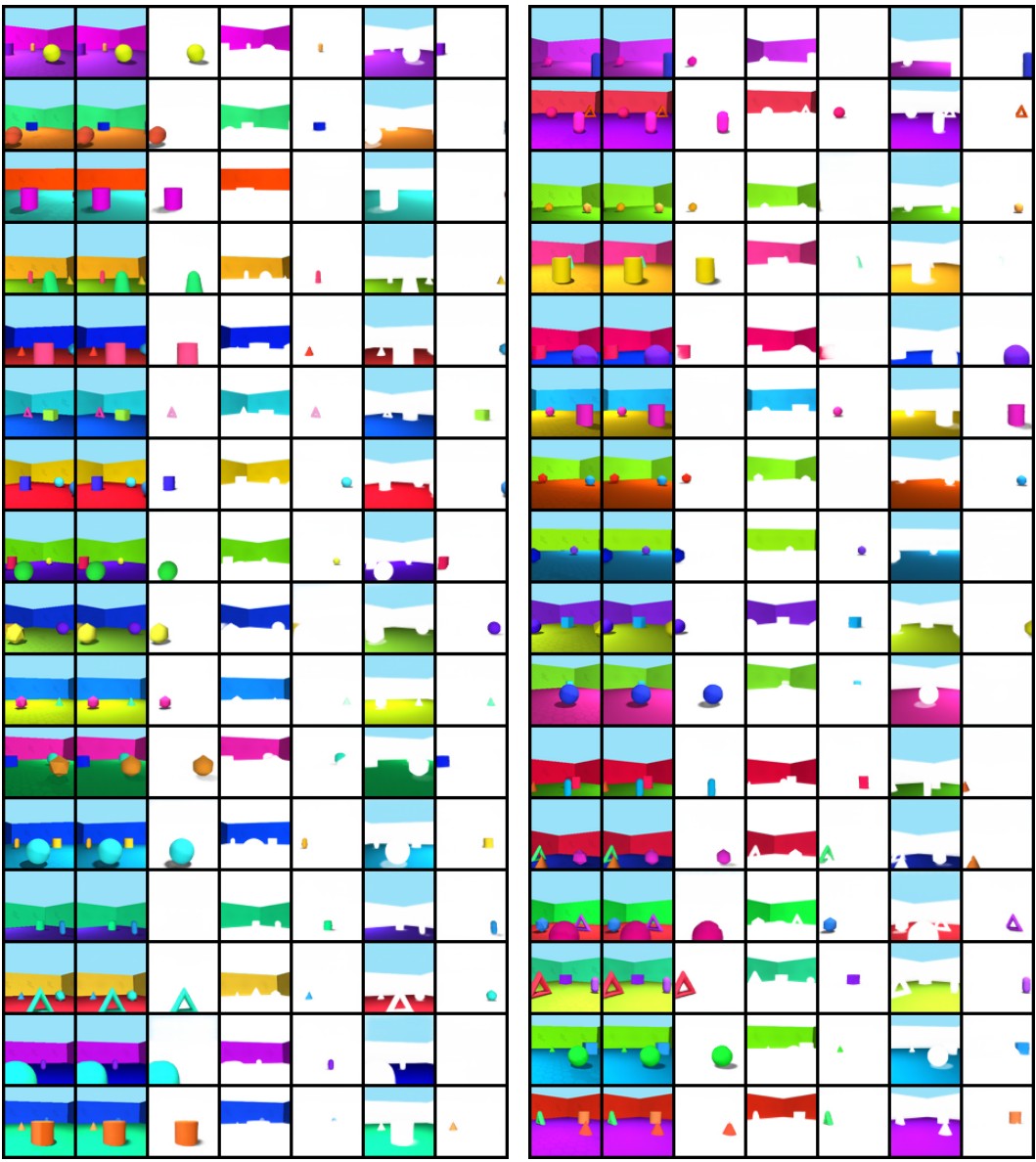

Figure 10: Unsupervised Multi-Object Segmentation on ObjectsRoom. In contrast to ShapeStacks, we observe consistent binding of slots to ground, wall, sky, and also objects in the front.

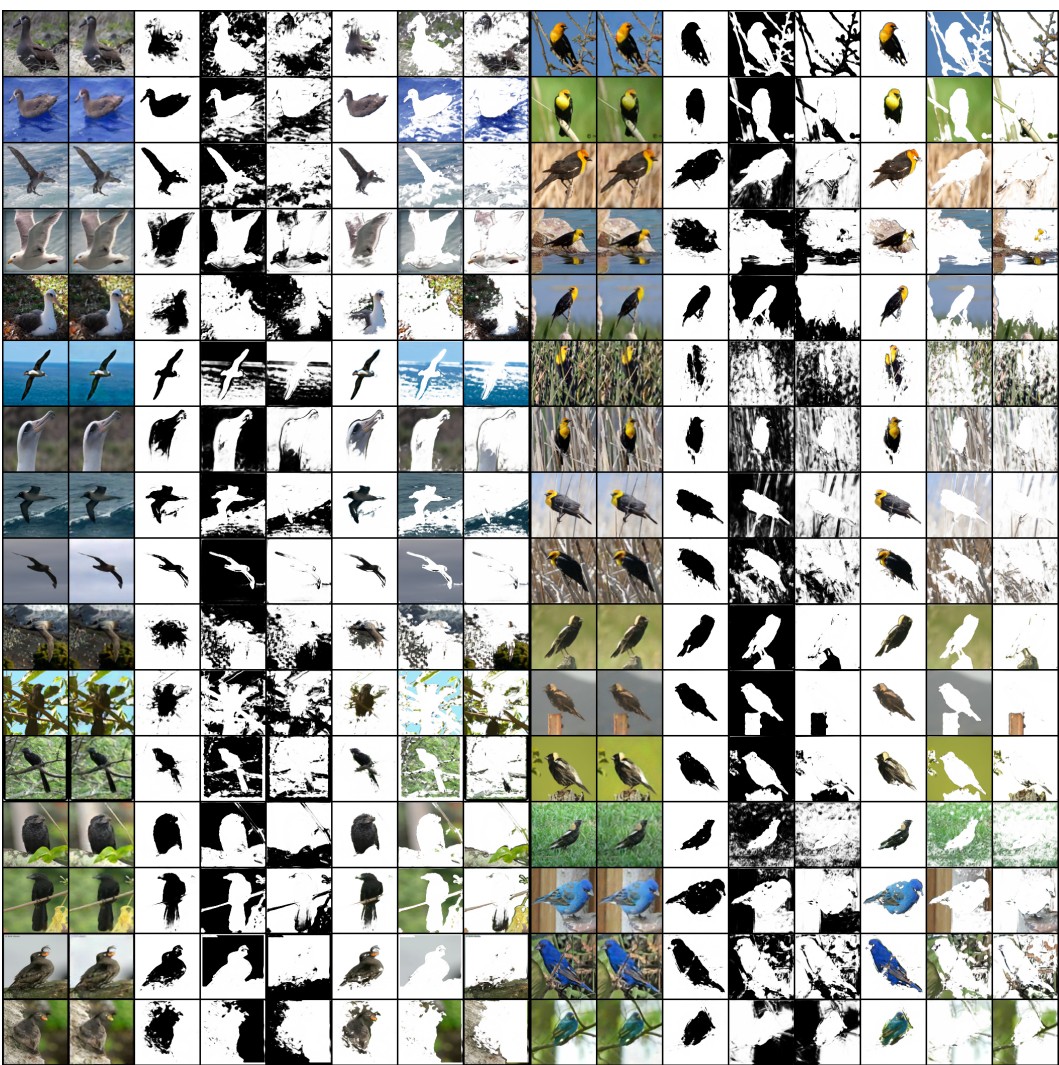

Figure 11: Unsupervised Foreground Extraction on CUB200 Birds.

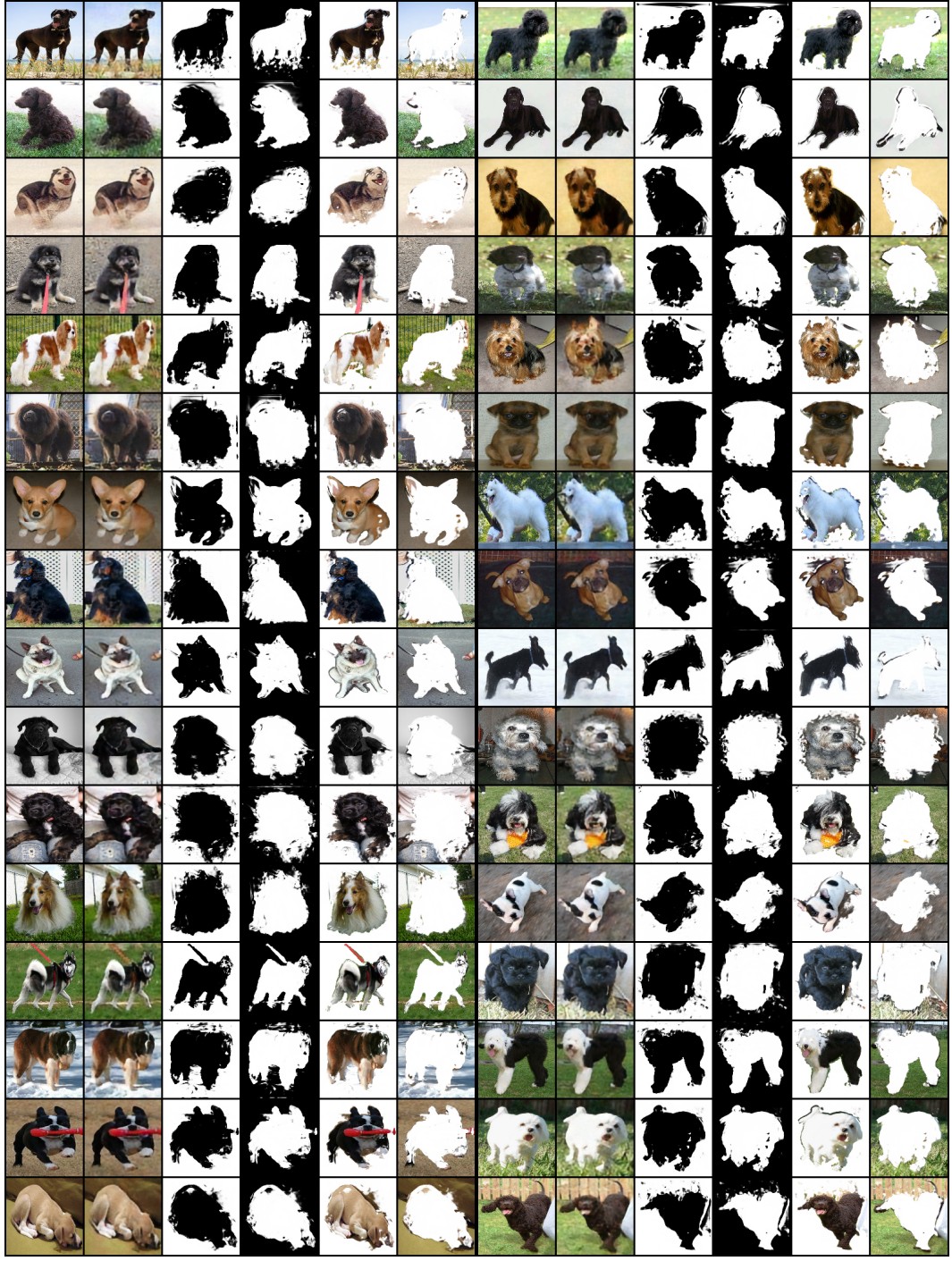

Figure 12: Unsupervised Foreground Extraction on Stanford Dogs.

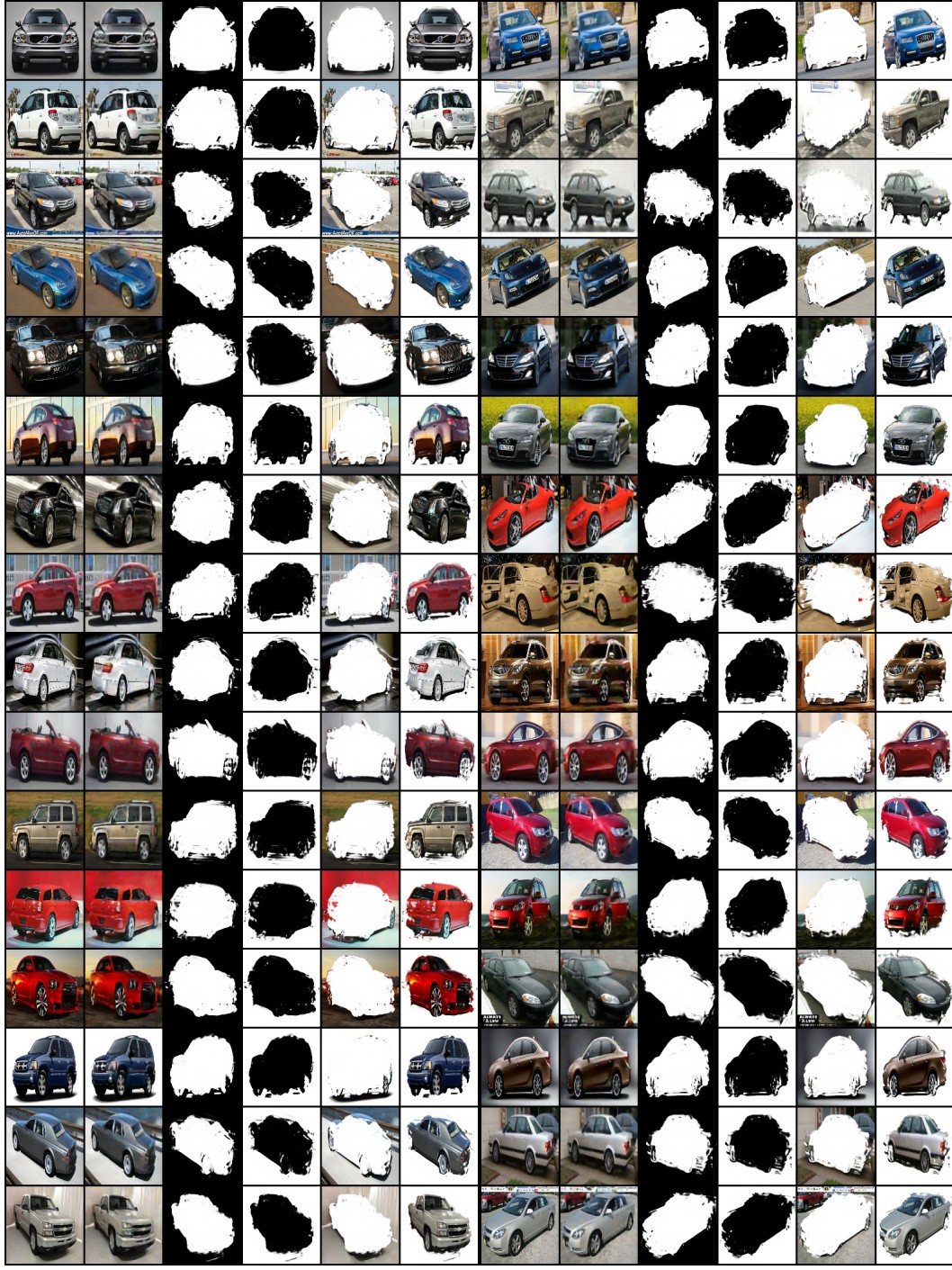

Figure 13: Unsupervised Foreground Extraction on Stanford Cars.

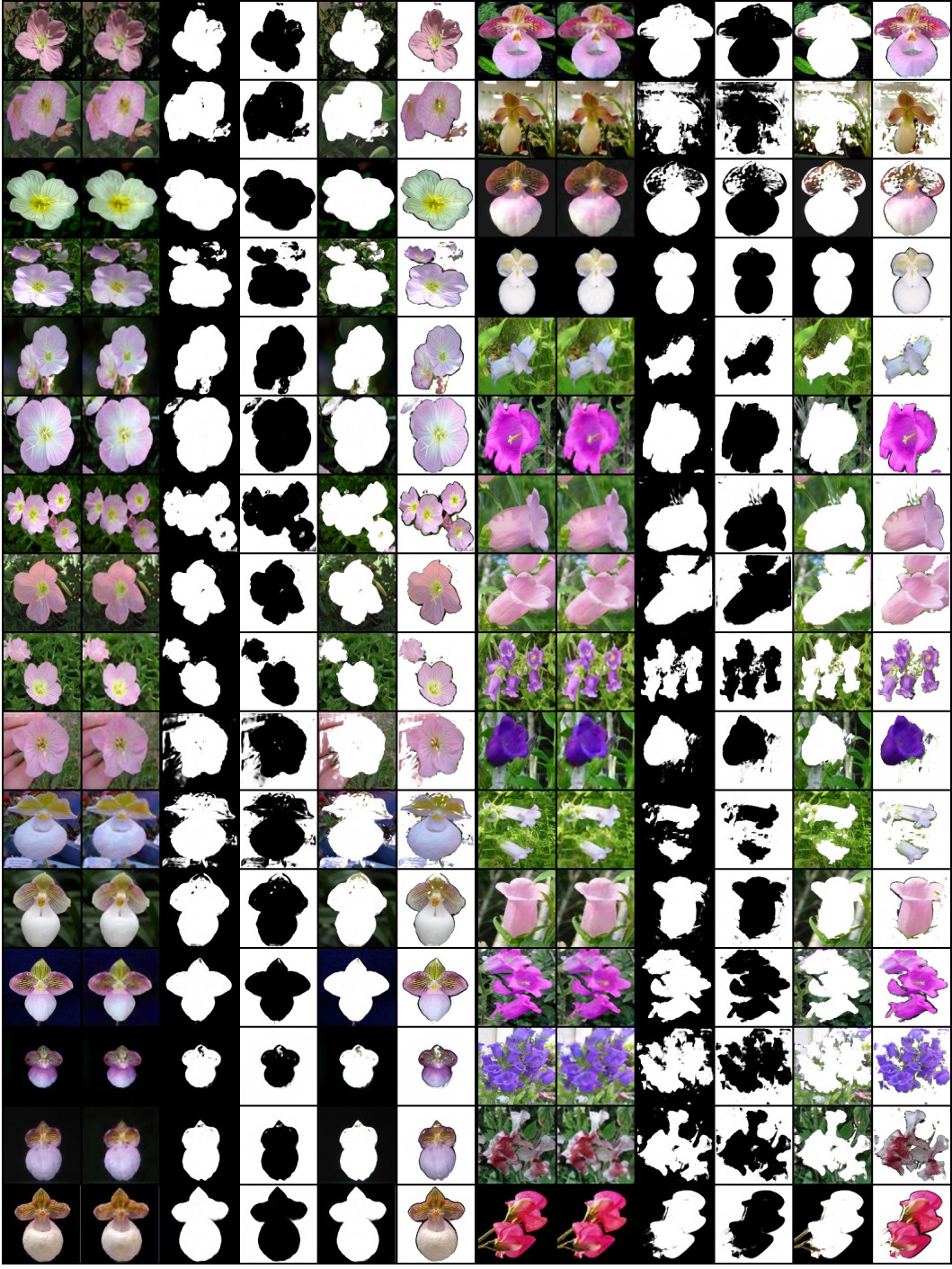

Figure 14: Unsupervised Foreground Extraction on Caltech Flowers.

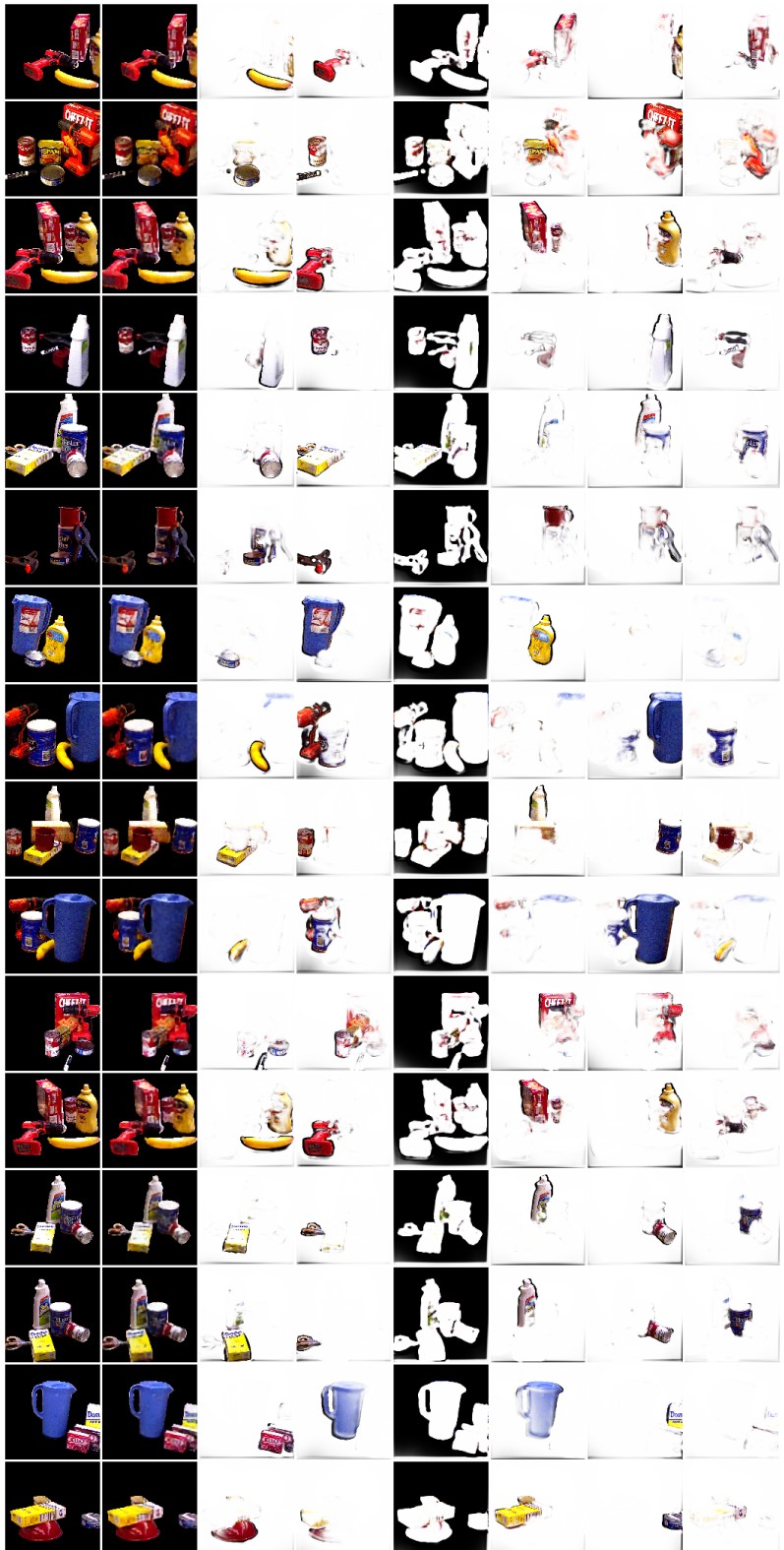

Figure 15: Unsupervised Multi-Object Segmentation on YCB.

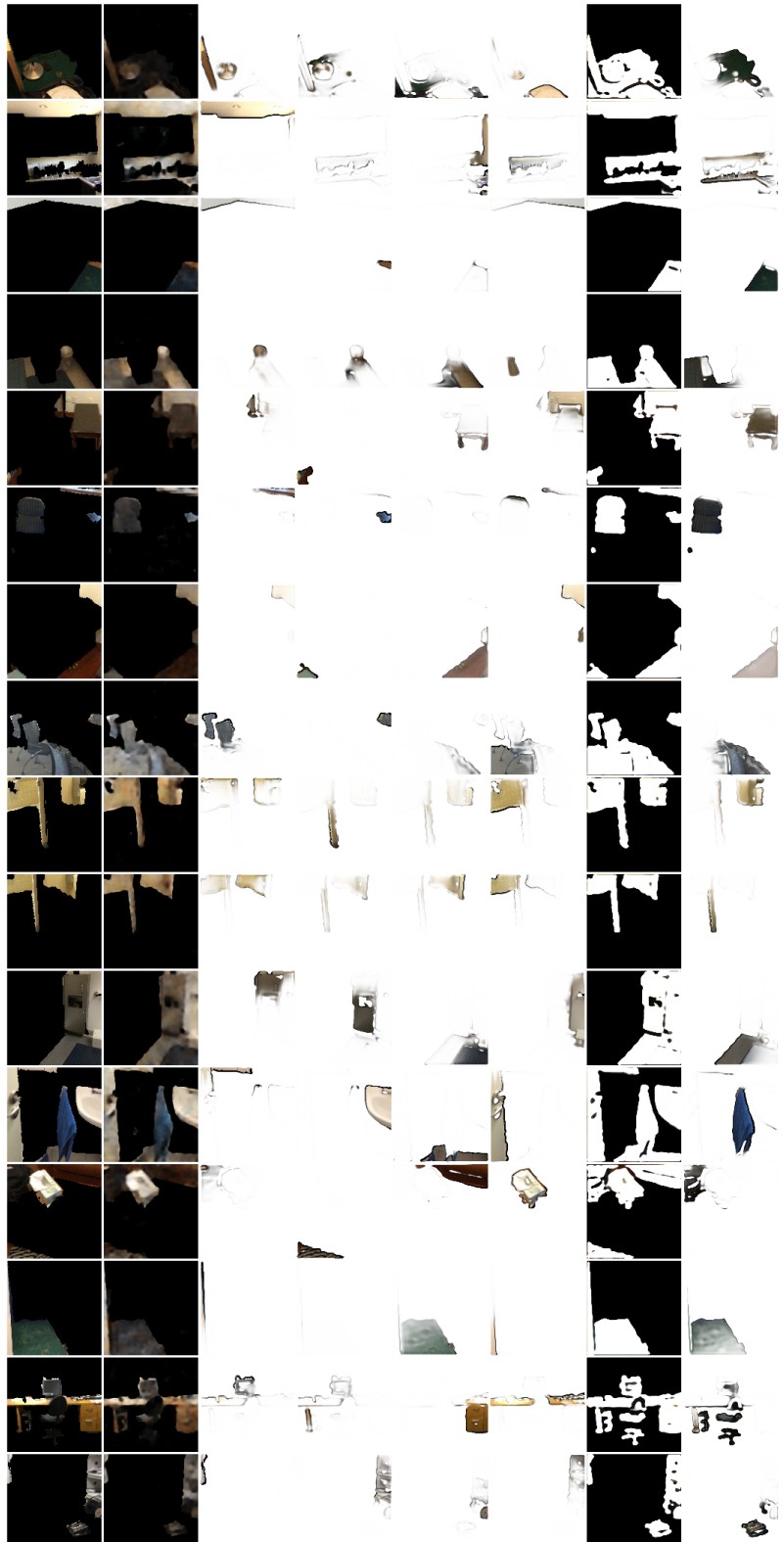

Figure 16: Unsupervised Multi-Object Segmentation on ScanNet.

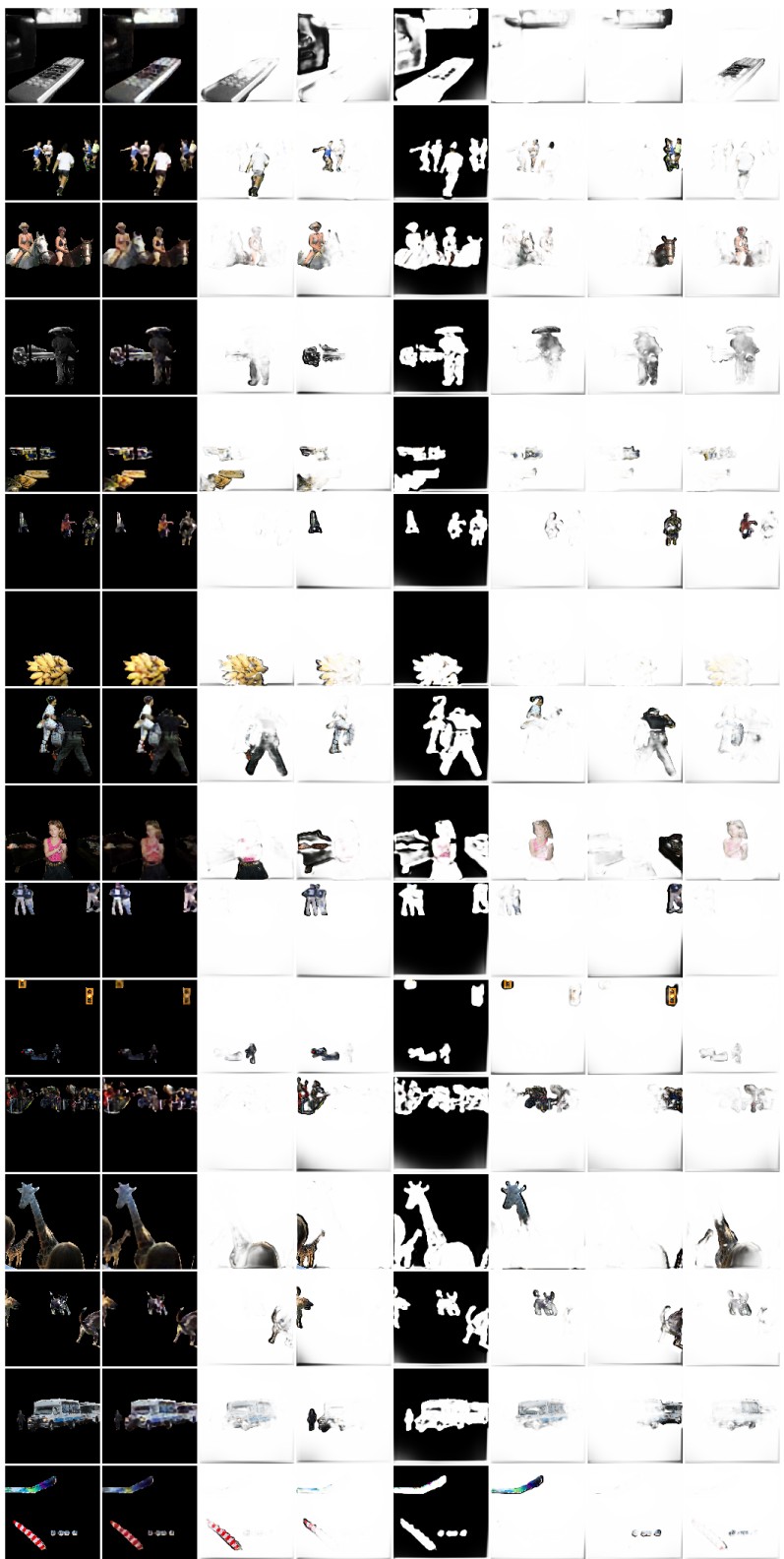

Figure 17: Unsupervised Multi-Object Segmentation on COCO.

