# OpenReview forum: "Improving Object-centric Learning with Query Optimization"
_ICLR.cc/2023/Conference — ICLR 2023 poster_

### Official Review · Reviewer_tD3C · 2022-10-20

**Confidence:** 4
**Correctness:** 4
**Technical Novelty And Significance:** 3
**Empirical Novelty And Significance:** 4
**Recommendation:** 6

**Clarity, Quality, Novelty And Reproducibility:**

The paper is written clearly and easy to follow. The implementation details are specific and the pseudo-code is tabulated in the paper, which benefits the reproducibility.

**Strength And Weaknesses:**

Strength:
+ The idea is neat and well-presented. Especially, table 5 clearly demonstrates the effectiveness of the method.
+ The two tricks complement each other, which strengthens the contribution of the paper. For instance, bi-level optimization makes the gradient backpropagate to slot initialization queries.
+ The method considerably boosts the performance on almost all the benchmarks.


Weakness:
+ The theoretical analysis is not sufficient. Despite the superior experimental results, the theoretical explanation is somewhat missing.
+ Learnable query is not novel. Previous work [1] has adopted this variant to stabilize the training.



[1] Self-supervised Video Object Segmentation by Motion Grouping. ICCV 2021.

**Summary Of The Paper:**

The paper proposes two tricks to optimize the training of slot attention. First, it initializes the query with learnable embedding instead of sampling from a learnable Gaussian distribution. Second, it applies bi-level optimization to the training. In practice, the slot binding process serves as the optimization of an inner optimization thus the gradient caused by this inner loop stops flowing backward. However, the learnable queries can still be updated with a straight-through estimator. The experiments on synthetic datasets (ShapeStacks and  ObjectsRoom) and real datasets (CUB200 Birds, Stanford Dogs, Stanford Cars, and Caltech Flowers) show that the proposed method achieves competitive performance. Moreover, the ablation study dissects the two tricks and proves both are indispensable.

**Summary Of The Review:**

In short, it is a good paper that takes simple tricks to boost the slot attention module. One possible improvement would be the theoretical explanation. Thus, I recommend accepting the paper.

---

> ### Author Response · Authors · 2022-11-18
> **Response to the Reviewer**
>
> We gratefully thank the reviewers for appreciating the simplicity and effectiveness of our proposed work. Please refer to the general response for a summary of all changes we have made in this revision. Here we provide further discussions on the reviewer's concerns:
>
> > Q1: More theoretical analyses
>
> As the reviewer requested, we revised the Sec.2 and Sec.3 of our manuscript to provide more analysis. First, the most important goal of object-centric learning is to obtain disentangled and expectable representations. The key to achieving this goal is finding controllable probes for learned concepts from models and grounding these learned representations. This is why we start with query-based Slot-Attention instead of sampling-based ones. However, as we discussed in Sec.3.2, ***existing methods (ISA) for improving sampling-based Slot-Attention training do not necessarily improve query-based Slot-Attention learning and even make it suffer***. This is why we looked into the bi-level optimization method and ***found that the straight-through gradient to the initialization queries is important since we can not assume the fixed number of Slot-Attention iteration guarantee the reaching of the optimal fixed point***. We also visualize this in Fig.3, proving this argument from the feature space visualization. We further validate this argument by showing that when improving the number of iterations during training, the I-QSA can improve (see Sec.B.2). Specifically, although the straight-through gradient estimator is biased, ***the learned initialization vectors can still help separate image features apart, and therefore leading to a good performance for object-centric evaluation***. Such a learned query initialization and encoder could serve the same purpose as other query networks learned with supervision(e.g., DETR), where we can query certain attributes of objects in the image that could generalize to new images (see Fig.4).
>
> > Q2: Minor issues
>
> We have added the necessary references mentioned by the reviewer in our revision. Thanks for the pointer. We hope the above response can resolve your questions and concerns. Please let us know if there is any further question!

---

> > ### Comment · Reviewer_tD3C · 2022-11-21
> > **Response to Rebuttal**
> >
> > Thanks for the authors' rebuttal. The revised paper reinforces the intuition of bi-level optimization in object-centric representation learning and the additional experiments make sense. And the visualization on multi-object data demonstrates that different slots could query certain attributes of objects. However, I agree with other reviewers that the transfer seems strange, e.g., the queries learned on birds can well transfer to cars. It seems contradictory to the intuition of learning slot initializations to query certain attributes, but only dividing the image into foreground and background parts. And the transfer on multi-object data is not that promising and cannot support the statement. Therefore, I would like to give marginally above acceptance.

---

> > > ### Author Response · Authors · 2022-11-22
> > > **Response to the reviewer (post-rebuttal)**
> > >
> > > We thank the reviewer for the constructive feedback. We want to emphasize on our main contributions that our major goal was to improve query-based slot-attention learning as it promises good potential. We proved our effectiveness through various experimetns (we believe 3 synthetic and 7 real is a lot). The concept binding effect comes as a side effect of the query-based slot-attention learning and we only claimed that they had good ***potential*** of such capabilities since the binding problem is not that well-defined without supervision. Therefore, our transfer experiments are there to further motivate query-based slot-attention methods (as an addtional study of our main experiments which already showed effectiveness). We list the quantitative transfer experiments here:
> > >
> > > |      Model      |      Slot-Attention     | Implicit Slot-Attention |             Ours            |
> > > |:---------------:|:-----------------------:|:-----------------------:|:---------------------------:|
> > > |  YCB to ScanNet |   1.37/4.90/11.27/6.35  | 21.62/21.81/32.32/34.19 | **28.24/25.93/36.68/39.62** |
> > > |   YCB to COCO   |   1.20/4.97/10.48/6.73  | 18.39/18.47/27.23/30.38 | **24.23/21.65/30.20/35.79** |
> > > |  ScanNet to YCB | 19.63/19.24/28.56/31.43 | 18.66/18.56/28.97/30.82 | **21.85/19.96/31.51/33.45** |
> > > | ScanNet to COCO | 12.84/14.86/22.06/26.74 | 11.83/14.14/20.70/25.42 | **13.95/16.04/23.35/28.49** |
> > > |   COCO to YCB   | 26.53/23.05/35.96/38.12 | 26.72/22.90/35.89/37.98 | **31.21/25.44/38.90/41.35** |
> > > | COCO to ScanNet | 20.99/22.08/32.14/36.53 | 19.34/20.00/29.44/33.18 | **24.21/23.59/34.07/38.49** |
> > >
> > > This is the most critical and direct evaluation on our method's transfer capabilities. The reviewer should notice that even for multi-object segmentation on real datasets, our transfer results are significantly outperforming previous works.
> > >
> > > As for the question you raised (on learning foreground background), we have tested on all foreground extraction datasets with different slot-attention variants and different slot numbers, all model configurations will tend to learn a rough contour of the foreground object as it is taking almost the entire image frame in those datasets (especially given that foreground objects tend to have unified colors, textures, etc.). This is why our model learned to segment the foreground as we hypothesize that the object attribute, foreground contours or textures, has been the clearest attribute to tell objects apart. Therefore, the correct way to interpret the current qualitative result is to check if the slots are learning the same set of attributes and can transfer to other dataset in the same data distribution (e.g. within the domain of synthetic, foreground extraction, or multi-object segmentation).
> > >
> > > If this analysis solves your concerns, we kindly ask the reviewer to reconsider the rating of our work.

---

> > > > ### Comment · Area_Chair_yXg2 · 2022-11-23
> > > > **Following up on rebuttal**
> > > >
> > > > Dear reviewer,
> > > >
> > > > you enjoyed the paper although you noted that novelty is not as high since learnable queries are well known. Has the response from the authors satisfied you?
> > > >
> > > > Cheers,
> > > > Your AC

---

> > > > > ### Comment · Reviewer_tD3C · 2022-11-23
> > > > > **Response to AC**
> > > > >
> > > > > Dear AC,
> > > > >
> > > > > Yes, I agree with the authors that this paper demonstrates the effectiveness of query slot attention from various aspects and this is a promising direction. But from my perspective, apart from the improvement on the quantitative performance, the concept binding is a significant attribute of query slot attention. Therefore, it is necessary to pay more attention to analysis into the phenomenon of concept binding and it might be promising leverage this attribute in training to further facilitate the qeury learning.

---

### Official Review · Reviewer_G5Uw · 2022-10-21

**Confidence:** 4
**Correctness:** 3
**Technical Novelty And Significance:** 3
**Empirical Novelty And Significance:** 3
**Recommendation:** 6

**Clarity, Quality, Novelty And Reproducibility:**

### General
- *Clarity*: The paper is in general easily understandable, and I see no major clarity issues besides the one mentioned in the weakness section.
- *Novelty*: Both design choices have been heavily inspired by previous work and done in some capacity. For example, the learning of slot initialization was already tested in the original Slot Attention paper, and other papers have investigated the iterative behavior of Slot Attention too. However, this paper does a good job in discussing them and provides a simple yet effective way of using them. To my knowledge, the combination of the design choices is novel.
- *Reproducibility*: While unfortunately not providing code, the method seems to be straight-forward to implement from Slot Attention. Hence, I would judge that the method should be reproducible. Nonetheless, I did not try to reproduce the method myself during the reviewing period.

### Minor points

- Table 3: it would be more consistent to add the citations of Slot Attention and SLATE to the model column, even if they had been cited before in the text.
- Appendix: The appendix seems to have swapped the caption below the tables instead of above

### Typos

- Page 2, Section 2.1 (second line): "a[n] iterative attention mechanism"
- Appendix Table 10: the word 'too' appears below DVAE
- Appendix Table 10: "Sof[t]max" is missing a "t"
- Appendix: Inconsistencies in writing "DVAE" vs "dVAE"



**Strength And Weaknesses:**

## Strengths

- The paper proposes two simple, but yet very significant improvements in Slot Attention considering the shown experimental results.
- The paper presents the two design choices clearly. In terms of re-producing the architecture, it requires changing only a few lines in the original Slot Attention model. Hence, the model could easily be adapted by many works in the field and be reproduced.
- To my understanding, the design choices do not introduce any additional hyperparameters and overall stabilizes Slot Attention. This is also important for future adaptations.
- The paper conducts extensive experiments on various datasets. I highly appreciated the variety of datasets, and it gives confidence that the method does not outperform Slot Attention just 'by luck' on a few cherry-picked settings.

## Weaknesses

- The paper spends a considerable amount of space on discussing the possible motivation behind the bi-level optimization, but it seems to me that the bi-level optimization is rather just an intuition on how one could connect the proposed setup to first-order meta-learning methods. For instance, what is the precise formulation of $\mathcal{L}_{\text{cluster}}$ that the model (implicitly) tries to solve? In Section 3.2, motivations are discussed for why it might be related to k-means, but it remains again a bit ambiguous. For the space used, I would have expected a clearer connection to the bi-level optimization.
- The paper misses a bit out on discussing the potential disadvantages of its method. For instance, by learning an initialization of the slots, the authors mention that they specialize to certain concepts. This would suggest that it may not work too well in out-of-distribution datasets that differ more in the 'concepts' learned. The zero-shot study is one step towards it, but the concepts remain quite similar in terms of foreground vs background. A more challenging situation is the out-of-distribution and CAMO evaluation datasets of CLEVRTEX. Since the authors already provide training results on the CLEVRTEX dataset, I believe it would be also important to report the evaluation scores for the OOD and CAMO part, even if the results may not outperform previous baselines.
- Additionally, since slots specialize on concepts, what happens if a test set image has a larger amount of objects of the same concept than seen in the training set, e.g. 10 red objects in ShapeStacks?

**Summary Of The Paper:**

This paper proposes two design choices to improve unsupervised object-centric representation learning of Slot Attention. The two design choices are to learn the initial slot embeddings instead of sampling them from a Gaussian distribution, and to skip the gradients through the iterative slot refinement by using a straight-through gradient estimator. This setup is extensively evaluated in experiments on unsupervised foreground segmentation, object-centric representation learning, and zero-shot transfer of foreground segmentation.

**Summary Of The Review:**

The proposed design choices are simple yet effective changes to a popular object-centric representation learning method. While easily understandable, the paper could be improved by having a clearer connection to its (intuitively) bi-level optimization objective and more discussions on out-of-distribution results. Overall, I think it is a good paper that is of interest to a larger community.

---

> ### Author Response · Authors · 2022-11-18
> **Response to the Reviewer**
>
> We thank the reviewer for acknowledging our experimental design and also the simplicity and effectiveness of our method. Please refer to the general response for a summary of all changes we have made in this revision. We further clarify the reviewer's concerns as follows:
>
> >Q1: The definition of inner objective and its connection to meta-learning
>
> We clarify the inner objective considered here and also rewrite section 3.2 accordingly. We treated slot-attention as a bi-level optimization problem where the outer objective solves reconstruction or set prediction, and the inner objective solves a clustering objective. More specifically, similar to K-means, we can write the clustering objective of slot-attention as follows:
>
> $$
> \mathcal{L}_{\text{cluster}}^{\text{Slot-Att}} = -\sum_p\sum_q\mathcal{K}_\Phi(x_i^p, s_q)
> $$
>
> $$
> \text{where}\ \mathcal{K}_{\Phi}(x_i^p, s_q) = [\mathcal{K}\_{\Phi}(\mathbf{x}\_i, \mathbf{s})]|\_{p,q} = \text{softmax}\left(\frac{k\_{\Phi}(\mathbf{x}\_i)\cdot q\_{\Phi}(\mathbf{s})^T}{\sqrt{D}}\right){\huge\vert}\_{p,q}
> $$
>
> We are looking for the "cluster centers" $\mathbf{s}$ that minimize this objective that best cluster per-pixel features ($x_i^p$) from image $\mathbf{x}_i$. The projections of the attention weights could be seen as a kernel $\mathcal{K}$ that maps input image features and centers to a latent space where we use the dot-product distance as our pseudo "distance measure". The reason why we relate this to meta-learning is to give a better picture and definition of task-specific parameters (poster-clustering slots in this case) and task-generalizable parameters (initializations and projections in this case). Moreover, to further motivate the need for query-initialized slot-attention modules. We believe the performance improvement from the query-initialized original Slot-Attention to our BO-QSA relies heavily on these adaptable and task-generalizable initialization queries. Our visualization of the effectiveness of initialization queries (see Fig.3) shows that ***compared with the sampling-based distribution, our initialization can better separate image features into clusters without ambiguous mixing after only one slot-attention iteration***. This also reflects our experiment on the number of iterations needed, as shown in Fig.2.
>
> >Q2: How will the model behave on more challenging data splits (CAMO/OOD in CLEVRTEX)
>
> First, we provide results on the CAMO/OOD in CLEVRTEX both here and in Tab.2.
> |      Model     |  CLEVRTEX-FULL  | CLEVRTEX-OOD | CLEVRTEX-CAMO |
> |:--------------:|:---------------:|:------------:|:-------------:|
> |      MONet     |   19.78(1.02)   |  37.29(1.04) |  31.52(0.87)  |
> |    Gensis-V2   |   31.19(12.41)  | 29.04(11.23) |  29.60(12.84) |
> | Slot-Attention |   62.40(2.33)   |  58.45(1.87) |  57.54(1.01)  |
> |       DTI      |   79.90(1.37)   |  73.67(0.98) |  **72.90(1.89)**  |
> | Ours (mixture) | **80.47(2.49)** |  **86.50(0.19)**            |       63.71(6.11)        |
>
> From what we observe, ***the current model consistently outperforms existing baselines except in CLEVRTEX-CAMO (though our model still outperforms slot-attention by a large margin)***. We attribute this phenomenon to the encoder used as we believe the current encoder (same as the original slot attention) is not powerful enough for learning useful features that define (separate) objects (Note that the current SOTA DTI considers stronger encoders with separate modeling for foreground and background).
>
> > Q3: What concepts are learned, and how will they behave in customized transfer settings
>
> Second, we clarify that **our learned concepts are sets of properties (attributes/axes) that define and separate objects**, not specific object concepts like dog, cat, or red cube. More intuitively, you can think of our slots binding toward objects with certain filters (e.g., color, position, contour, etc.). We show in the revised version of our paper (see Fig.4) that with more objects of the same concept existing (e.g., two or more same-colored objects), the slots will filter out all objects that belong to that concept. This is also reflected by the OOD results in CLEVRTEX, where our model kept showing superior performance since we are learning "descriptors" of objects for clustering instead of specific object concepts. Such learned concepts fit our need for image/scene interpretations as they could potentially be descriptors with higher-level semantics (compared with traditional edge or color filters). Although we can not control the properties learned currently, this also proves the potential of our slot initializations to bind toward desired concepts with additional supervision (e.g., weak supervision from language).
>
> >Q4: Minor issues
>
> We have fixed references in all tables, captions in the appendix, and typos in our manuscript's revision. Thanks for pointing them out.
>
> We hope the above response can resolve your questions and concerns. Please let us know if there is any further question!

---

> > ### Comment · Area_Chair_yXg2 · 2022-11-23
> > **Following up on rebuttal**
> >
> > Dear reviewer,
> >
> > similar to reviewer HyQy, you ask authors to clarify on the motivation and in the conundrum of specialization (better modelling with specific number of objects) vs generalization (worse modelling but with any number of objects). Has the response influenced your views?
> >
> > Cheers, Your AC

---

> > > ### Comment · Reviewer_G5Uw · 2022-11-27
> > > **Rebuttal response**
> > >
> > > I would like to thank the authors for their rebuttal and addressing my concerns. The additional experiments strengthen the empirical part of the paper and show its application to e.g. CLEVRTEX. However, I do agree with reviewer HyQy that the theoretical contribution of this paper, especially with respect to the bi-level optimization arguments, are still not fully convincing. Considering all together, I would like to keep my initial rating of Weak Accept.

---

### Official Review · Reviewer_HyQy · 2022-10-24

**Confidence:** 4
**Correctness:** 2
**Technical Novelty And Significance:** 2
**Empirical Novelty And Significance:** 3
**Recommendation:** 3

**Clarity, Quality, Novelty And Reproducibility:**

Quality: low. The proposed approach is a marginal modification of existing methods. Rather than focusing on reporting higher numbers on tasks such as foreground/background segmentation, I would have preferred a more thorough analysis of the theoretical implications of connecting bi-level optimization and slot-based learning. Instead, the connection feels artificial and unjustified, also considering that the optimization used in practice is not bi-level.

Clarity: medium. The paper is well written, both the textual parts and the math notation. The only disconnect point is section 3.2, which happens to be the most important point of the paper where the connection between bi-level optimization and slot attention should be made explicit.

Originality: low. Both implicit slot optimization and learnable query tokens are not new ideas. The connection with bi-level optimization would make this paper original, but I find it rather weak and poorly explored.

Reproducibility: high. Implementation details are given in the main text and in the appendix.

**Strength And Weaknesses:**

Strengths:

- In terms of implementation, the paper proposes two straightforward design changes on top of regular slot attention: 1) initial slots are learned rather than sampled as in Locatello 2020, 2) gradient updates are skipped in intermediate steps as Chang 2022 and are propagated to the initial slots using a straight-through estimator.

- I appreciate the thorough experiments which cover both synthetic and real datasets and different decoder designs. The proposed method achieves higher scores on all datasets considered, which is a point in its favor.

Weaknesses:

- In a side-by-side comparison with the implementation of "Object representations as fixed points: Training iterative refinement algorithms with implicit differentiation" (Chang 2022), this paper makes only one slight change in the gradient propagation. Specifically, gradients from the last step of the clustering loop are used to update the initial slots (straight-through estimator). This is a quite minor change, which may be seen as an ablation study of Chang 2022. Apart from the empirical experiments that show a performance improvement in some benchmarks, which I discuss below, I don't see any interesting theoretical insight related to this change. Sure, there is a lengthy introduction on bi-level optimization in section 3.2, but it feels artificial and disconnected from the practical change proposed in section 3.3. If this paper studies the connection between slot attention, k-means clustering, and bi-level optimization, what theoretical insights can be drawn from this previously unexplored connection? I think this is the main question that the paper should answer rather than focusing on the empirical results.

- Since the main contribution of this work is the bi-level optimization of the initial query vectors, I expected more experiments studying the effect of the bi-level optimization on the final performance. What are the implication of propagating gradients to the initial query vectors? In particular, does the straight-through estimator introduce a bias in the gradients? How different are the learned queries from the post-clustering slots? Do the intermediate clustering steps become redundant due to the straight-through estimator, essentially turning this method in a VQ-VAE? How would other methods for learning initial queries behave, e.g. a running mean of the post-clustering slots or some actual bi-level update mechanism? I think these are interesting questions that the paper should address.

- In Locatello 2020, the original slot attention is introduced as soft unsupervised clustering method that can be initialized with a variable number of random vectors. One of the most interesting experiments was the ability to train on images containing a certain number of objects and generalizing to a larger number of objects simply by increasing the number of sampled slots. Learning the slot initialization precludes this possibility because the number of learnable queries must be set as a hyperparameter, and also requires a certain a priori domain knowledge. Discussion of this drawback is lacking in the paper, instead, empirical results are presented on synthetic data where the number of object is known and on foreground/background segmentation where two slots suffice.

- The choice of how many slots to use for each dataset is not made explicit in the main text (only in the appendix). Setting the number of slots to 2 for most foreground/background tasks is very convenient but does not convince about the generality of the method. For the birds dataset it is set to 3 for some unclear reason.

- Two points regarding the ablation studies:
  - First, ablation studies usually focus on hyperparameter choices of the proposed method, e.g. number of slots or how to optimize them. Instead the ablation studies presented in the paper are a comparison with previous methods. This comparison should arise from the main experiments, not from the ablation studies.
  - Second, I am confused by some combinations of slot initialization and optimization procedure. Why were these two combinations chosen?
    - I-QSA: in QSA the initial slots are initialized at random and then learned, but they receive no gradient due to the stop-gradient operation, which makes this combination equivalent to I-SA. Unless there are other differences, I would attribute the lower performance of I-QSA wrt I-SA to an unfortunate initialization.
    - BO-SA: in SA the slots are sampled at random so propagating gradients to them (BO) should have no effect and should be equivalent to I-SA. The results are in fact very similar.

- Section 5.4 the statement that BO-QSA has the potential to be used for learning concepts is overstated in my opinion. The only supporting evidence is that slots tend to specialize on colors for ShapeStacks in figure 3. However, this behavior has been observed in other slot-based methods too, and can be explained simply by noting that even a traditional pixel clustering algorithm would capture RGB colors. I would not draw any conclusion about the ability to learn concepts from these color-based observations. The other example given in figure 3 uses the birds dataset where the number of slots has been conveniently set to 3 for foreground/background segmentation. Since the birds dataset contains images of a single subject over a blurry textured background, it's rather obvious that the slots will specialize on the subject and the background. This is not a very convincing example of concept learning.

- Related to the point above, the conclusion also contains an overly bold statement "By further verifying our findings under zero-shot transfer learning settings, our model shows great potential as a principle approach for learning generalizable object-centric representations". Though I acknowledge the transfer learning results, I would highlight that the experiment in question is foreground/background segmentation on rather simple datasets. I would not extrapolate these observations to truly multi-object real-world datasets like COCO, LVIS, or PASCAL.

Minor points:

- Why the connection with meta-learning? Bi-level optimization has many applications and meta-learning is just one of them. Nowhere else in the paper the connection with meta-learning is leveraged to justify or enhance the proposed method. I think the entire section 2.2 could be removed without hindering the message of the paper.

- Appendix A.1, check the paragraph names

**Summary Of The Paper:**

The paper draws a connection between slot attention and bi-level optimization.
As a consequence, the authors suggest to learn the initial query slots rather than sampling them as in previous work. Furthermore, the design choice of Chang 2022 w.r.t. gradient propagation is adopted and validated.
Empirical results show the potential of this method for unsupervised instance segmentation on synthetic datasets and foreground/background segmentation on real-world datasets.


**Summary Of The Review:**

On the first pass through the paper, I was initially excited to read about the connection between bi-level optimization and slot attention, also thanks to the higher reported scores on several benchmarks. However, I then realized that this connection is not well-motivated and does not add any new insight. The message of the paper could have been simply "learn the query vectors with a straight-through estimator".

The paper is well written and the authors conducted experiments on several datasets, but I think the paper does not contribute novel knowledge. Surely, the empirical numbers are higher, but the choice of experiments is not on point and does not justify acceptance. I recommend rejecting the paper, but I remain open to changing my mind upon reading the rebuttal and discussing with the other reviewers.

---

> ### Author Response · Authors · 2022-11-18
> **Response to the Reviewer (4/4)**
>
>
> > Q5: Are concepts learned? Is it better than previous models?
>
> We respectfully disagree with the reviewer on the notion that "traditional color pixel clustering algorithm could capture RGB colors" and believe it is not a fair comparison to our methods given that ***our encoder did not add any inductive bias on colors and is learned end-to-end solely by reconstruction***. The high sensitivity to colors is not a trivial outcome of unsupervised learning methods, especially given that all compared baselines discussed in this paper have not yet established such stable binding to object color properties. This is also the same for foreground extraction given the transfer results in Sec.B.1. We also show that our binding towards object properties is not limited to color and contours but also spatial positions in Fig.4. As the emergence of such concepts rely solely on learning the data distribution without inductive bias, we believe it does show the potential of our model to successfully bind concepts with proper supervisory signals (e.g. with weak language alignment supervision).
>
> > Q6: More convincing transfer settings
>
> We have added experiments on recent real-world unsupervised multi-object segmentation datasets mentioned by reviewer 9uz4. We show that ***our model's transfer capability is not limited to foreground/background segmentation and is consistently outperforming current baselines*** by a large margin in Tab.8 (Sec.5) and Tab.14 (Sec.B.1). We also provide the transfer results here:
>
> |      Model      |      Slot-Attention     | Implicit Slot-Attention |             Ours            |
> |:---------------:|:-----------------------:|:-----------------------:|:---------------------------:|
> |  YCB to ScanNet |   1.37/4.90/11.27/6.35  | 21.62/21.81/32.32/34.19 | **28.24/25.93/36.68/39.62** |
> |   YCB to COCO   |   1.20/4.97/10.48/6.73  | 18.39/18.47/27.23/30.38 | **24.23/21.65/30.20/35.79** |
> |  ScanNet to YCB | 19.63/19.24/28.56/31.43 | 18.66/18.56/28.97/30.82 | **21.85/19.96/31.51/33.45** |
> | ScanNet to COCO | 12.84/14.86/22.06/26.74 | 11.83/14.14/20.70/25.42 | **13.95/16.04/23.35/28.49** |
> |   COCO to YCB   | 26.53/23.05/35.96/38.12 | 26.72/22.90/35.89/37.98 | **31.21/25.44/38.90/41.35** |
> | COCO to ScanNet | 20.99/22.08/32.14/36.53 | 19.34/20.00/29.44/33.18 | **24.21/23.59/34.07/38.49** |
>
> > Q7: Writing & logical issues
>
> We thank the reviewer's comment on the logical flow of this paper. We indeed feel that the mentioned problems affected the clear understanding of our contribution. Therefore, we have followed the reviewer's suggestion and moved the discussion of connections between our method and meta-learning into the appendix. This paragraph was mainly aimed at providing a good intuition on why our model could provide good transfer results, as both the learned initialization queries could be seen as object property filters that share across datasets and provide useful information for segmentation. We have also modified Sec.2.2, Sec.3.2, and Sec.3.3 such that ***we clarify in Sec.2.2 why our current method is already a bi-level optimization method, motivate the need for a straight-through estimator in Sec.3.2 and describe our model in Sec.3.3***.
>
> > Q8: The paper does not contribute novel knowledge, empiricial numbers are higher but not to the point
>
> We humbly point out that our design is, quoted from reviewer g5uw, showing the effectiveness of a new combination of existing modules (from other tasks) in the task of unsupervised object-centric learning. We believe our extensive experiments help to illustrate the motivation for our designs, highlight how such a method could benefit existing methods in an extremely straightforward fashion, and are sufficient for validating our claims and did our best at addressing the reviewer's concerns. ***Together with Chang 2022, we position our work as addressing an existing bottleneck for slot-attention and object-centric learning, which paves the way for learning disentangled representations and concepts on structured and noisy natural data, and can be efficiently adapted to other object-centric applications. We sincerely hope such well-motivated, simple, and effective efforts should not be neglected.***
>
> We hope the above response can resolve your questions and concerns. Please let us know if there is any further question!

---

> > ### Comment · Area_Chair_yXg2 · 2022-11-23
> > **Following up on rebuttal**
> >
> > Dear reviewer,
> >
> > the authors tried to address the points in your review, most notably on the motivation and in the conundrum of specialization (better modelling with specific number of objects) vs generalization (worse modelling but with any number of objects). Do you agree or has the response influenced your views?
> >
> > Cheers, Your AC

---

> > > ### Comment · Reviewer_HyQy · 2022-11-25
> > > **Post-rebuttal comments**
> > >
> > > Dear authors, thanks for the very extensive rebuttal.
> > >
> > > You have added many experiments that strengthen the empirical results. I surely appreciate that. At the same time, I am not convinced of the contribution of this work besides better numbers. Specifically, I am failing to see the importance of bi-level optimization which is supposed to be central to the paper (it's in the title and in the name of the method). From the way the paper is written, I find it hard to frame the contributions as either very theoretical or very empirical. It seemed very theoretical in the initial version and instead it focuses on empirical results in the post-rebuttal version.
> > >
> > > As an experiment, I tried to remove all mentions of bi-level optimization from the paper. Essentially, there is no major difference in the message of the paper: "use learned queries for slot attention to achieve higher scores in many benchmarks". This is what I meant in the original review when complaining that no new insight is derived by the previously unexplored connection with bi-level optimization. The paper can stand on its own even without this theoretical connection.
> > >
> > > Also, the authors motivate using learned queries vs sampled queries by saying that the former leads to "disentangled and expectable representations" while the latter does not. I argue that this is neither necessary nor guaranteed:
> > > - In SLATE, which is used as a comparison in the paper, the cluster centers are randomly initialized. Yet it's possible to collect and visualize a library of visual concepts. Therefore, learned queries are not really necessary for interpretability.
> > > - Regarding expectability, the method does not seem to guarantee that. Figure 4 shows that the object-centric criterion learned for one dataset (e.g. cluster by color for ShapeStacks) does not guarantee object-centric representations on other datasets (cfr. CLEVRTEX). Is the behavior predictable? Sure, things are clustered by color. Does the method learn to separate object instances? Not really.
> > > - I'm also confused about the claim of disentanglement. In fact, there is no experiment that proves that specific dimensions in the learned representations are associated with specific attributes of the captured objects. To be clear, my idea of disentanglement is that of "beta-VAE: Learning Basic Visual Concepts with a Constrained Variational Framework" (Higgins et al. 2016) and subsequent works, where changing one element in the vector changes one attribute of the output e.g. color, size, or shape. This is not to be confused with object-centric learning where the learned representations are associated with distinct objects but do not necessarily disentangle their attributes.
> > >
> > > Overall, my opinion of this paper is not fully negative. The empirical results are good and the experiments are extensive, at least in the post-rebuttal version. At the same time, I do not think that this paper contributes much knowledge to the field of object-centric learning: the connection with bi-level optimization is unnecessary and no new insight is drawn from it, the learned queries yield expectable results that however correlate more with color/position than with a notion of objectness. All considered I'm open to increasing my recommendation to 5 "marginally below the acceptance threshold" but I would still recommend rewriting this paper and resubmitting elsewhere.

---

> > > > ### Author Response · Authors · 2022-11-26
> > > > **Response to the reviewer (post-rebuttal)**
> > > >
> > > > We thank the reviewer for reading through our rebuttal and revised paper. However, we strongly disagree with the reviewer's statement that "hard to frame the contribution as either very theoretical or very empirical". We have run experiments for **3** synthetic (including PTR then it's **4**) and **7** real-world datasets. We observed consistent performance advantage of the proposed model over previous ones (**oftentimes by a large margin**), and to our knowledge, this is the first model that shows such capabilities on such a wide range of tasks especially on challenging new real-world datasets proposed. This contributes to the first and major contribution of the paper and the reviewer could definitely treat it as an empirical one that should not be overlooked.
> > > >
> > > > As for the bi-level optimization, we have **followed the reviewer's suggestion and listed the motivations of our designs to revise our paper**, especially on (1) **how bi-level optimization is crucial for the optimization slot attention (Sec.2.2 and 3.2)** (Chang et al. 2022); (2) **how the same bi-level optimization technique do not fit well for query-based learning (Sec.3.2, 3.3, B.2)**; and (3) **how we leveraged the STE to backpropagate the gradient to queries to improve query-based slot attention together with bi-level optimization**. Our ablation study in Tab.7 shows that without the bi-level optimization technique discussed in Sec.2.2, where we use Neumann's series to approximate Eq.(3) (i.e., detaching the previous slots and running one-step of slot-attention to backpropagate gradients to attention projection, gru, etc.), both slot-attention and query slot attention is struggling on datasets. We also provide additional experiments in Sec. B.2 and B.3 to show that (i) **compared to other initialization methods our STE learned queries perform the best**, and (ii) **the bi-level optimization method proposed by Chang et al. faces the problem of inaccurate fixed point approximation, but, as an extension of their work, our method can improve the fixed point approximation**. These experiments and analyses should have delivered the message that neither of these two methods are dispensable.
> > > >
> > > > Finally, for the learned concepts, we first correct the reviewer's impression of SLATE where the authors of **SLATE used K-means on post-clustering slots to collect a library of visual concepts**. This is essentially different from ours as **we do not perform any post-clustering procedures and each slot can already show potential in binding to certain concepts**. As discussed in our responses to the reviewer and other reviewers, in our experiments our slots **showed potential** in binding attributes, this is a result of using a fixed-number of queries since the number of objects increased but the number of slots didn't. For such scenarios, we argue that the current result is intuitive and desired as we can categorize objects by attributes. The other thing that should not be missed is that we conducted zero-shot transfer experiments on all real-world dataset combinations and our method showed better results. We believe with these results, our contribution, **showing the potential** for concept binding, is not over-claimed, especially given that we treated it as additional analyses instead of a major experiment. The reviewer might as well provide suggestions on how we phrase the contribution of the transfer experiments and binding certain concepts to slots.
> > > >
> > > > Above all, we hope these additional comments help the reviewer see our contributions better. We are also open to any follow-up discussions.

---

> > > > > ### Author Response · Authors · 2022-12-02
> > > > > **Response to the reviewer (post-rebuttal) follow-up**
> > > > >
> > > > > We want to thank you again for your time and constructive comments. We hope that we have addressed your concerns in the previous discussions and are open to any follow-up discussions. If there are no further concerns, we kindly ask you to reconsider the rating of our paper, especially given all positive comments from other reviewers. We believe our work would promote future research in this field.

---

> ### Author Response · Authors · 2022-11-18
> **Response to the Reviewer (3/4)**
>
> > Q3: Generalizing to more slots, potentials in query-based slot-attention
>
> We respectfully disagree with the reviewer on the view of query slot attention. As mentioned by other reviewers, object-centric learning with learnable queries is, in fact, a widely studied topic as they could be optimized to be having expectable behaviors and fit for controllable probing of concepts. ***The balance between generalization to an additional number of objects and the controllableness of each output is a natural outcome between using a fixed number of learnable parameters versus sampling from a distribution***. With slot-attention showing capability in generalizing to more object scenarios, it fails at attending concepts to the same queries with its randomness. It is also possible for slot attention to start with similar initializations (see Fig.3) and generate coarse object segmentations (e.g. uniformly distributing background or objects into slots). We emphasize again that our work aims at addressing the potential of slot-attention module as a controllable probe for concepts that supports future research towards more general intelligence. Therefore, we argue this comparison on generalizing to more slots is not on point, especially with our model showintg better separation of input features (see Fig.3) and also learning properties that could filter out concepts across datasets (see Fig.4). As the reviewer suggested, we have added another limitation section in Sec.6 and Sec.C to make the advantages and disadvantages of our model clearer to the readers.
>
> > Q4: Choice of number of slots
>
> We remind the reviewer that the design of Slot-Attention was originally for multi-object segmentation and required proper hyperparameter tuning on the number of slots used as it is a critical factor. For the experiments on foreground extraction, since the Slot-Attention was not designed for this task, ***previous methods also tuned this hyperparameter and reported the best results***. We followed the same protocol (used in [1]) to find the best slot number configurations for our model on each dataset similar to previous evaluations and therefore does not produce any unfairness between the evaluation of models.
>
> [1] Yu et al., "Unsupervised foreground extraction via deep region competition", NeurIPS 2021
>
> > Q5: Clarifications of the ablation study
>
> We kindly remind the reviewer that we ran all experiments on the same model configuration and only compared the usage of initialization, detach, and straight-through gradient estimator. With our BO-QSA taking all three designs, the ISA, SA, QSA, I-QSA, BO-SA are only ablative variants of our models with different combinations of the three modules used and are therefore, ablative models. Next, we clarify the details of ablative experiments.
> - In the I-QSA, we used query vectors to initialize slot attention and detached the slots as described in Chang 2022. ***This method leads to a query-based slot attention with no direct gradient back to the learnable query initializers***. To attribute for its low performance compared with I-SA, the reviewer made a key observation that without gradient and sampling, the learnable queries could be wrongly initialized and lead to imperfect fixed point solutions that result in low-performance results. This is exactly the reason why we want to add straight-through gradient estimation to adjust these initializations to better learn from the data.
> - For BO-SA, ***the straight-through gradient estimator will backpropagate gradients into the mean $\mu$ and variance $\sigma$ that defines the initialization distribution***. This is different from I-SA as it detaches slots from the first few iterations completely and does not backpropagate gradients to $\mu$ and $\sigma$. Though the effects are similar on real-world images, the gap between them on Shapestakcs is not negligible.

---

> ### Author Response · Authors · 2022-11-18
> **Response to the Reviewer (2/4)**
>
> > Q2: The effect of the proposed initialization methods, how is it different from VQ-VAE or directly using the post-clustering slots
>
> We appreciate the reviewer's key observation on the connection between slot initializations and post-clustering slots. We have added a discussion on this issue in Sec.B.3. Here, we provide a discussion on the use of running-mean of post-clustering slots or adding constraints between the two using VQ-VAE losses. During the initial experiments, ***we tried (1) updating with post-clusering means, (2) reclustering post-clustering slots and updating initialization vectors by Hungarian matching to these reclustered centers***. We provide results both here and in Tab.17. To improve stability, we also consider using momentum updates for these two variants.
>
> |         Model        | ShapeStacks (ARI-FG) | ShapeStacks (MSC-FG) |
> |:--------------------:|:--------------------:|:--------------------:|
> |      RunningMean     |          7.5         |          3.7         |
> | RunningMean-Momentum |         51.4         |         15.4         |
> |        KMeans        |          1.2         |          3.1         |
> |    KMeans-Momentum   |         70.6         |         60.4         |
> |     VQ-constraint    |         88.6         |         85.3         |
> |         Ours         |         **92.9**         |         **89.2**         |
>
> ***All of these methods lead to uniform slots that maintain the mean-image of inputs, thus showing inferior performance***. We found a similar phenomenon for adding the VQ-VAE consistency loss ($|sg(\hat{s}) - s_0|^2$) as the initialization queries are learned to be mean statistics of input images and are not beneficial for separating objects (see in Fig.6). In contrary, our learned initializers are trained to separate different objects each time without the constraints on its distribution being similar to input feature distribution. They could therefore be powerful disentanglers instead of cluster centers that, when combined with the attention mechanism, lead to a good separation of image features.

---

> ### Author Response · Authors · 2022-11-18
> **Response to the Reviewer (1/4)**
>
> We thank the reviewer for the clear summary of our proposed models and also for acknowledging that our experiments are thoroughly designed, showing our improvement is, as the other reviewers stated, not achieved just "by luck". However, we respectfully disagree with the reviewer simply summarizing our paper as "learning query vectors with straight-through estimators," as it is a critical intermediate step toward the goal of learning disentangled and expectable representations and also a non-trivial problem to solve under the slot attention model (even with previous works improving it).  Please refer to the general response for a summary of all changes we have made in this revision. We make further clarifications to address the reviewer's concerns as follows:
>
> > Q1: What is the motivation of the proposed learned slot initializations, what new theoretical insights, why bi-level optimization
>
> First, we emphasize that our ultimate goal of this work is to obtain an object-centric model with disentangled and expectable representations. We aim to learn unsupervised models that we hope can always preserve the same set of filtered properties (concepts) when applying to new images. This is not only important for the problem of clustering or segmentation but also the basis for future research on concept learning, language grounding, etc. ***This is the key motivation why we focus on query-based architectures since sampling-based initializations forbid this potential***. However, as shown by previous works and also our experiments, such queries do not work well with slot-attention training. Considering it as an optimization problem, we need to backpropagate gradients into the iterated fixed point operation $F_{\Phi}$, and this causes unstable gradients to both the weights in slot-attention and the learned initialization queries. ***This is the reason why we look into bi-level optimization methods, especially with the inner problems solved by iterative updates***. We show that Chang 2022's results are not enough for query-based slot attention (as shown in Tab.5 and Fig.3). This is the major difference between our work and Chang 2022 and why we are not ablation studies of their work.  We show that it is crucial to add a gradient path toward the slot initializations by showing (1) when improving the number of iterations during training, the I-QSA can improve (see Sec.B.2), and (2) as shown in Fig.3, our initializations can create a better separation between slots and attended features. (1) suggests that a fixed number of iterations do not guarantee to obtain the optimal fixed point and (2) suggests that by learning the initialization queries, our model indeed exhibits better potential. Although the straight-through gradient estimator is biased, the learned initializations can still help separate image features apart, accounting for the good performance of object-centric evaluation. Such a learned query initialization and encoder could serve the same purpose as other query networks (e.g., DETR) where we can query certain relationships in the image that could generalize to new images.

---

### Official Review · Reviewer_9uz4 · 2022-10-25

**Confidence:** 4
**Correctness:** 3
**Technical Novelty And Significance:** 3
**Empirical Novelty And Significance:** 3
**Recommendation:** 6

**Clarity, Quality, Novelty And Reproducibility:**

This paper has a clear and complete structure and is easy to read. The implementation details and experiment settings are also very clear for reproduction. However, this paper is a bit like an extension based on Slot Attention and Implicit Slot Attention (ISA). Therefore its novelty is somewhat discounted.

**Strength And Weaknesses:**

Strength:

1. The structure of the paper is very clear. The motivation and related research are well-explained. The technical details are concise.

2. The proposed module is simple and effective, which could serve as a plug-and-play module for many models.

3. It explicitly learns object concepts from datasets as slot initializations, which paves the way for object-centric learning in more challenging cases.


Weakness:

1. More insights on learned slot initializations are expected:

Figure 3 has shown slot contents for a given input image after iterative updating. But what’s more special in this paper is the learned slot initializations, which are shared among all images in the dataset. It is expected that the learned slot initializations are the object concepts abstracted from the dataset. You may convert the learned slot initializations as images for visualization, or perform feature space analysis such as T-SNE to provide more insights.

2. More discussions on the effectiveness of new design are needed:

Although ablation experiments in Section 5.3 have demonstrated the effectiveness of the proposed module, it is still unclear why learnable queries as slot initializations can bring these large improvements. You may include more analysis and comparison with SlotAtt especially in the experiment part. For example, SlotAtt mentions that its typical solution is to distribute the background equally over all slots. However, from the visualization of this paper, it seems all background pixels are separately assigned into a single slot. A more detailed and analytical comparison with SlotAtt are needed to provide more insights on your contribution.

3. More challenging experiments are expected:

The experiment parts demonstrate remarkable results on a set of synthetic and real datasets. But for all synthetic datasets including those in the appendix, the objects are simple-colored and mostly simple-shaped, where a color-based bias may already work very well. It is suggested to evaluate on more complex synthetic datasets such as ClevrTex [1].

For the real datasets, all images only have a single foreground object, and the task becomes a simple binary classification. It’s more convincing to evaluate on multi-object real images, as also discussed in a very recent paper [2].

[1] ClevrTex: A Texture-Rich Benchmark for Unsupervised Multi-Object Segmentation, NeurIPS 2021.
[2] Promising or Elusive? Unsupervised Object Segmentation from Real-world Single Images, NeurIPS 2022.

4. Zero-shot transfer learning experiments:

Successful zero-shot transfer learning experiments suggest generalizable representations are learned. However, if a slot learned on dog dataset can be easily transferred to flower dataset. Does it imply the slots are not necessarily binding with object concept, at least not a specific type of object? You need to investigate what object concept is learned. Are they objects such as cat and dog, or a set of properties
that defines objects?

**Summary Of The Paper:**

This paper proposes an extension of SlotAtt (NeurIPS 2020) by adopting learnable queries as slot initializations. The learning process is formulated as a bi-level optimization problem, where image reconstruction is the outer objective and soft clustering of image feature is the inner objective. In order to stabilize the training procedure, this paper detaches gradients to the recursive updates, only keeping gradients for last iteration and slot initialization queries.

The proposed method is evaluated for unsupervised object discovery task on synthetic datasets and unsupervised foreground extraction task on real datasets. It demonstrates improved performance on both tasks compared with state-of-art unsupervised object-centric models, especially the vanilla SlotAtt.

Additionally, this paper shows the potential to bind object concepts to its learned slots with experiments on zero-shot transfer learning.

**Summary Of The Review:**

This paper is a simple but effective extension of Slot Attention. The structure and presentation are great. It demonstrates remarkable improvements on unsupervised object discovery and foreground extraction. However, the major concerns are: 1) deeper technical insights/analysis/discussion are missing, 2) more convincing experiments are needed.

Consider the simple and effective design and the excellent performance, the reviwer is happy to increase the score if the concerns can be well addressed or explained.

---

> ### Author Response · Authors · 2022-11-18
> **Response to the Reviewer (2/2)**
>
>
> > Q3. Results on more challenging datasets
>
> We thank the reviewer for the pointer on the recent real-world multi-object segmentation work and include these challenging datasets in our revision. For synthetic datasets, we provide the results of our model (with mixture decoder as it's our best configuration for synthetic datasets) on all splits of ClevrTex here and in Tab.2.
>
> |      Model     |  CLEVRTEX-FULL  | CLEVRTEX-OOD | CLEVRTEX-CAMO |
> |:--------------:|:---------------:|:------------:|:-------------:|
> |      MONet     |   19.78(1.02)   |  37.29(1.04) |  31.52(0.87)  |
> |    Gensis-V2   |   31.19(12.41)  | 29.04(11.23) |  29.60(12.84) |
> | Slot-Attention |   62.40(2.33)   |  58.45(1.87) |  57.54(1.01)  |
> |       DTI      |   79.90(1.37)   |  73.67(0.98) |  **72.90(1.89)**  |
> | Ours (mixture) | **80.47(2.49)** |  **86.50(0.19)**            |       63.71(6.11)        |
>
> We observe that ***the current model consistently outperforms existing baselines except in CLEVRTEX-Camo (though our model still outperforms the slot-attention by a large margin)***. We attribute this phenomenon to the encoder used as the current encoder (same as the original slot attention) is not powerful enough for learning useful features with camouflaged objects. The current SOTA method DTI uses a stronger encoder that makes separate modeling for foreground and background objects. For real-world experiments, we follow the instructions in the reference mentioned and conducted three trials of experiments on all three datasets (YCB, ScanNet, and COCO). We show that without additional data modifications, ***our model (with transformer decoder as it is our best configuration on real-world images) significantly outperforms existing baselines on all three benchmarks***, as shown here and in Tab.4.
>
> |        Model       |                 YCB (AP/PQ/Pre/Rec)                 |               ScanNet (AP/PQ/Pre/Rec)               |                 COCO (AP/PQ/Pre/Rec)                |
> |:------------------:|:---------------------------------------------------:|:---------------------------------------------------:|:---------------------------------------------------:|
> |         AIR        |         0.0(0.1)/0.6(0.3)/1.1(0.4)/0.8(0.2)         |         2.7(1.4)/6.3(1.7)/15.6(2.8)/7.3(1.6)        |         2.7(0.1)/6.7(0.5)/14.3(2.6)/8.6(0.8)        |
> |        MONet       |         3.1(1.6)/7.0(2.6)/9.8(3.6)/1.2 (0.8)        |       24.8(1.6)/24.6(1.6)/31.0(1.6)/40.7(1.8)       |       11.8(2.0)/12.5(1.1)/16.1(0.9)/21.9(1.7)       |
> |       IODINE       |         1.8(0.2)/3.9(1.3)/6.2(2.0)/7.3(1.9)         |       10.1(2.9)/13.7(2.7)/18.6(4.2)/24.4(3.8)       |         4.0(1.2)/6.3(1.2)/9.9(1.8)/10.8(2.0)        |
> |   Slot-Attention   |        9.2(0.4)/13.5(0.9)/20.0(1.3)/26.2(6.8)       |        5.7(0.3)/9.0(1.5)/12.4(2.5)/18.3(2.7)        |         0.8(0.3)/3.5(1.2)/5.3(1.7)/7.3(2.2)         |
> | Ours (transformer) | **48.0(1.8)/34.8(1.3)/50.8(1.1)/53.6(0.7)** | **28.5(2.4)/26.4(2.0)/37.3(2.0)/42.4(1.9)** | **17.8(0.6)/17.6(0.6)/25.3(0.6)/30.6(0.9)** |
>
> We believe these experiments can sufficiently show that our work, as commented by reviewers G5Uw and tD3C, did not outperform previous methods just 'by luck' and could be used as a powerful design for object-centric learning.
>
> We hope the above response can resolve your questions and concerns. Please let us know if there is any further question!

---

> > ### Comment · Area_Chair_yXg2 · 2022-11-23
> > **Following up on rebuttal**
> >
> > Dear reviewer,
> >
> > the authors tried to respond to your points regarding insights of the learned slots, as well as the complexity of experiments. Are you satisfied, or you think the results are not enough at the moment?
> >
> > Cheers,
> > Your AC

---

> > > ### Author Response · Authors · 2022-11-30
> > > **Rebuttal follow-up**
> > >
> > > Please feel free to post additional questions and comments about our work during the author-reviewer discussion period, we are happy to discuss and continue improving our works during this phase. Or, if we have successfully addressed most of your concerns, we would appreciate it if you kindly consider raising your final rating.

---

> > > > ### Comment · Reviewer_9uz4 · 2022-12-01
> > > > **Update**
> > > >
> > > > Dear authors,
> > > >
> > > > Thanks for the very detailed response and additional explanation and experiments, which together makes the current version much better than before. I'd be happy to improve my score. One more suggestion:
> > > >
> > > > Quantitive companions with the following (concurrent) method are desired, as this approach shares some spirits with it in terms of parameter optimisation strategy if I understand correctly.
> > > >
> > > > -- Object representations as fixed points: Training iterative refinement algorithms with implicit differentiation, NeurIPS 2022

---

> ### Author Response · Authors · 2022-11-18
> **Response to the Reviewer (1/2)**
>
> We gratefully thank the reviewer for acknowledging the simplicity and effectiveness of our proposed model. Please refer to the general response for a summary of all changes we have made in this revision. We further make the following clarifications to address the reviewer's concerns:
>
> > Q1: Insights on learned slot initializations and their connection to concept binding:
>
> We appreciate the reviewer's key observation on the connection between object initializations and learned concepts. Importantly, we want to clarify that **our learned concepts are sets of properties (attributes/axes) that define and separate objects**. We show in the new experiments (see Fig.4) that our learned slot initializations bind to not only color but also spatial position and object contours. We visualize the attention map between slot initialization queries and images on transferred new datasets (see Fig.4). As we can see from the figure: these learned initializations can be further used as concept filters for color, position, or contours. Although we can not control which properties to learn currently, this proves the potential of our slot initializations to bind toward desired concepts with additional supervision (e.g., weak supervision from language).
>
> > Q2. Discussions on the effectiveness of the new design
>
> To elaborate on the role of our new design in the soft-clustering process, we visualize the slot initializations, as well as the slot update vectors after the first iteration of slot attention. As we can see from Fig.3, ***our slot initialization queries can better separate image contents for each slot compared with vanilla sample-based slot attention, leading to better segmentation results and significantly reducing the number of ambiguous segmentations*** (e.g., distributing background into different slots). We further compare this visualization with other variants (see Fig.3) and show that our learning strategy for slot initializations is essential for good performance.

---

### Author Response · Authors · 2022-11-18
**General response to the reviewers**

We follow comments and suggestions from all reviewers and revise our manuscript (colored in blue) accordingly as follows:

- 1. We clarify the goal of our work on obtaining better object-centric models with potential in binding concepts. To this end, we improve ***query-based Slot-Attention*** learning with both bi-level optimization and straight-through gradient estimators. We have revised Sec.2.2 and Sec.3, moving less-related discussion on meta-learning to Sec.A.2 and making descriptions of objectives clear [**reviewer HyQy,G5Uw**].
- 2. We add more theoretical discussion and analysis on (1) the ineffectiveness of ISA for query-based Slot-Attention in Sec.3.2 and (2)the effectiveness of our initialization in Sec.3.3 and Sec. 5.3 with t-SNE visualization of learned initializations. We also provide a comparison between different ways for updating initialization queries in Sec.B.3, addressing the concerns of [**reviewer 9uz4**,**HyQy**,**tD3C**].
- 3. We add experimental results on more challenging datasets with realistic images, including all splits on CLEVRTEX, YCB, COCO, and ScanNet. We also add zero-shot transfer experiments on real multi-object segmentation datasets (YCB, COCO, ScanNet) in Sec.5.4, Sec.B.1). Overall, our model consistently achieves the ***best*** results, and ***outperforms previous methods by a large margin***. [**reviewer 9uz4**, **HyQy**, **G5Uw**]
- 4. We provide additional qualitative results of learned concepts and their effect when applying the learned initialization queries to other datasets in Sec.5.3. In Fig.4, we show that our model learns attributes (color, contour, spatial location) that define objects and provide visualization of hypothetical scenarios mentioned by the reviewers for verification (e.g., multiple objects with the same color) [**reviewer 9uz4**, **G5Uw**]
- 5. We include the discussion of limitations and future works in both Sec.6 and Sec.C. [**reviewer HyQy**, **G5Uw**]

We appreciate all the suggestions made by reviewers to improve our work. It is our pleasure to hear your feedback, and we look forward to answering your follow-up questions.

Paper2269 Authors

---

### Decision · Program_Chairs · 2023-01-20

**Decision:**

Accept: poster

**Justification For Why Not Higher Score:**

No clear theoretical motivation, despite the very interesting good results.

**Justification For Why Not Lower Score:**

The results are quite positive and all reviewers agreed on that, even the negative one.

**Metareview: Summary, Strengths And Weaknesses:**

The paper proposes the use of learned slot initialization, as well as bi-level optimisation to improve object-centric learning. Experimental results are strong, with extensive experiments in synthetic and read data. All reviewers acknowledge the empirical findings and their importance. Also, all reviewers think that the theoretical contributions are not that clear, and the paper overclaims.

This paper was certainly a borderline case, details  and suggestions in the next question.

**Note From Pc:**

if the above contains the word "oral" or "spotlight" please see: "oral" presentation means -> notable-top-5% and "spotlight" means -> notable-top-25%. As stated in our emails, we are disassociating presentation type from AC recommendations

**Summary Of Ac-Reviewer Meeting:**

This paper was certainly a borderline case: one strong reject vs three weak accepts. What is more, all reviewers agreed on all points. The main difference is whether to weigh more the empirical findings over clarity of motivation and theoretical contribution.

In the AC-reviewer meeting, we discussed whether the empirical findings are (1) significant enough, and (2) whether there is any reason to believe that the numbers are not plausible. All reviewers agreed that the numbers are plausible, in fact one of the reviewers partially reimplemented the method and seem to give consistent results.

The main criticism is whether the current focus on bi-level optimization obscures and distracts, giving the wrong interpretation. We all agreed on this point, and that likely improvements are due to a practical change in the way slots are defined (learned, instead of sampled at initialization), and not what complex story the paper currently tries to promote.

The question then was whether the changes were too big to warrant a resubmission. In a journal format, this submission would likely be a 'Major revision'. In a conference format, we do not have such an option. As progress in our field is not always in complete sync with complete and perfect understanding (eg residual connections), and empirical/practical findings can fuel and encourage further research, I concluded that the paper should be given the benefit of the doubt and be accepted. For sure, 3 out of 4 experienced reviewers find the practical message relevant.

However, the acceptance is conditional. The current title, abstract, and main text relies too heavily on aspects that are not what lead to the improvements most likely. Thus, the authors are asked that for the camera-ready the change the title, abstract to not emphasize on bi-level optimization (a better title/abstract can instead emphasize the learned slots), and modify the text to also tone down the contribution of the bi-level optimization.

---

> ### Author Response · Authors · 2023-02-10
> **Camera-ready revision**
>
> We followed the reviewers' suggestions and changed the title/abstract to focus on "Query Optimization", acknowledging Bilevel optimization as an existing framework for improving Slot-attention learning. Meanwhile, we toned down on the discussion of Bilevel optimization, mainly identifying its potential problems with initialization to motivate the design of query learning/optimization. At last, we express our greatest gratitude to all reviewers for their valuable suggestions on improving our work.